# Using Cosmic-Ray Neutron Probes in Validating Satellite Soil Moisture Products and Land Surface Models

**Mustafa Berk Duygu and Zuhal Akyürek *** 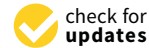

Water Resources Laboratory of Civil Engineering Department, Middle East Technical University,
06800 Ankara, Turkey
* Correspondence: zakyurek@metu.edu.tr; Tel.: +90-312-210-2481

**Abstract:** Soil moisture content is one of the most important parameters of hydrological studies. Cosmic-ray neutron sensing is a promising proximal soil moisture sensing technique at intermediate scale and high temporal resolution. In this study, we validate satellite soil moisture products for the period of March 2015 and December 2018 by using several existing Cosmic Ray Neutron Probe (CRNP) stations of the COSMOS database and a CRNP station that was installed in the south part of Turkey in October 2016. Soil moisture values, which were inferred from the CRNP station in Turkey, are also validated using a time domain reflectometer (TDR) installed at the same location and soil water content values obtained from a land surface model (Noah LSM) at various depths (0.1 m, 0.3 m, 0.6 m and 1.0 m). The CRNP has a very good correlation with TDR where both measurements show consistent changes in soil moisture due to storm events. Satellite soil moisture products obtained from the Soil Moisture and Ocean Salinity (SMOS), the METOP-A/B Advanced Scatterometer (ASCAT), Soil Moisture Active Passive (SMAP), Advanced Microwave Scanning Radiometer 2 (AMSR2), Climate Change Initiative (CCI) and a global land surface model Global Land Data Assimilation System (GLDAS) are compared with the soil moisture values obtained from CRNP stations. Coefficient of determination ($r^2$) and unbiased root mean square error (ubRMSE) are used as the statistical measures. Triple Collocation (TC) was also performed by considering soil moisture values obtained from different soil moisture products and the CRNPs. The validation results are mainly influenced by the location of the sensor and the soil moisture retrieval algorithm of satellite products. The SMAP surface product produces the highest correlations and lowest errors especially in semi-arid areas whereas the ASCAT product provides better results in vegetated areas. Both global and local land surface models' outputs are highly compatible with the CRNP soil moisture values.

**Keywords:** Cosmic Ray Neutron Probe; land surface model; SMAP; SMOS; AMSR2; ASCAT; CCI; GLDAS

## 1. Introduction

Soil water content is one of the most influential variables that is used in the decision support systems of land and water management studies. However, in order to utilize soil moisture data, it must have an applicable spatial and temporal resolution. Measurement of soil moisture is possible via several methods [1] including laboratory tests (requires intensive labor) and time domain reflectometers (TDR) (having high accuracies but smaller measurement footprints), ground penetrating radar [2] and remote sensing methods (having large measurement footprints but lower accuracies and resolutions), proximal gamma ray spectroscopy sensing [3,4] and Cosmic Ray Neutron Probes (CRNPs) [5] (having intermediate horizontal scale compared to the other methods). The footprint of a CRNP is a circle

with a 670 m diameter as suggested by Reference [5] or a 360–420 m diameter as suggested by Reference [6]. Point scale soil moisture observations can provide insight into small areas which are generally much smaller than the study area of any comprehensive study while satellite soil moisture products cover much larger areas but their spatial resolution is questionable and most of them do not provide continuous data. In order to fill the gap between spatial and temporal resolutions of these two types of soil moisture products, ground measurements that have intermediate spatial resolution constitute an important potential. CRNPs can be used in satellite products' validation and to improve the resolution and the accuracy of operational satellite soil moisture products as well as calibrating land surface models, which use satellite information as the input parameters with the help of their intermediate scale horizontal footprint. Error characterization of soil moisture products is important because of the following reasons: (i) the result of the validation can be used as feedback for algorithm developers for further improvements in the retrieval of soil moisture; (ii) to facilitate understanding of the status of the product for potential users, such as the accuracy, magnitude and the uncertainties of the remote sensing products. Both are very important as they help in better understanding their potential use for practical applications like numerical weather prediction, rainfall estimation, flood forecasting, drought monitoring and prediction [7].

There are several studies which compare satellite soil moisture products with CRNP stations in the literature [8–12]. These studies mostly focus on the correlation between a satellite soil moisture product and a CRNP measurement or average values of a group of CRNP stations by investigating the coefficient of determination and triple collocation (TC) errors. Studies with passive sensors investigate the difference between ascending and descending products and they point out that the ascending products better represent the ground measurements [13]. A similar study conducted in 2017 compares various space-borne soil moisture products (SMOS, ASCAT, AMSR2, SMAP, GLDAS) which are similar to those used in this study with sixteen different CRNP stations located at six different basins and the study suggests that SMAP surface soil moisture product gives similar results with CRNP's [14]. A recent study investigated the possible usage of active and passive satellite soil moisture products (SMAP, ASCAT and AMSR2) in a combined manner and suggests that a combined dataset that is obtained by using SMAP and ASCAT better represents ground observations [10]. In another study, ASCAT soil moisture product was evaluated at different sites globally representative of a variety of climatic, environmental, biome, and topographical conditions [15]. Although no definite conclusion was reached regarding the ubRMSE performance over different seasons, regarding the Pearson's correlation coefficient, the ASCAT data was found to perform better in autumn and winter than in spring and summer.

The aim of this study is to assess the use of CRNP soil moisture data in the validation of different types of satellite soil moisture products (active or passive products at ascending and descending nodes), modeled soil moisture as well as a combined soil moisture dataset (CCI). The main motivation of this study is to extend the validation study [14] to all CRNPs of COSMOS database with expanding the period of validation and to enhance the study [15] with presenting the use of CRNPs in validating the satellite soil moisture products. Validation of satellite products is carried out with data obtained from CRNPs at different locations in the world including Çakıt Basin in Turkey. In addition, validation studies are enhanced with a land surface model forced by local data and in-situ TDR measurements at Çakıt Basin.

## 2. Data and Methodology

### 2.1. Satellite and Land Surface Model Based Soil Moisture Products

In this study, four different satellite products, which are namely METOP-A/B Advanced Scatterometer (ASCAT), Soil Moisture and Ocean Salinity (SMOS), Soil Moisture Active and Passive (SMAP), Advanced Microwave Scanning Radiometer (AMSR2) and a global land surface model Global Land Data Assimilation System (GLDAS), were used.

According to Reference [13], passive microwave remote sensing products are more successful for their morning passes. In order to test this property of passive remote sensing products, comparisons of SMOS and AMSR2 were conducted for both ascending and descending node products as well as their average values.

Information related to selected satellite based soil moisture products and their corresponding grid cell representing the Çakıt Basin CRNP station are shown in Table 1. Stations in the COSMOS database were also compared with the satellite products at the closest grids to their locations. In this study, soil moisture values obtained from both satellite grids and the CRNPs were assumed to have homogeneous distributions over the areas, thus their values were directly compared with each other. For this reason, satellite products were not spatially or temporally rescaled and were used as they are without filling any missing values since filling the missing data by using adjacent grids or time points may adversely affect the analyses .

**Table 1.** Satellite soil moisture products that are compared with Çakıt Station COsmic Ray Neutron Probe (CRNP) soil moisture content data.

| Soil Moisture Product | Pixel Size (km) | Coordinate of the Pixel Center (Closest to Çakıt CRNP) |
|---|---|---|
| METOP-A/B Advanced Scatterometer (ASCAT) EUMETSAT H113-H114 SSM | 12.5 × 12.5 | 37.597° N 34.625° E |
| Soil Moisture and Ocean Salinity (SMOS) L3 1-day Binned Product | 25 × 25 | 37.482° N 34.484° E |
| Soil Moisture Active and Passive (SMAP) L4 3-hourly EASE-Grid SSM | 9 × 9 | 37.4746° N 34.4969° E |
| Advanced Microwave Scanning Radiometer (AMSR2) L3 1 day c band 6.9 Ghz | 9 × 9 | 37.55° N 34.45° E |
| ESA Climate Change Initiative (CCI) v04.4 (Active, Passive and Combined) | 25 × 25 | 37.5155° N 34.4979° E |
| Global Land Data Assimilation System (GLDAS) Noah LSM L4 3 hourly V2.1 | 25 × 25 | 37.625° N 34.375° E |

### 2.1.1. METOP-A/B Advanced Scatterometer (ASCAT)

ASCAT refers to the advanced scatterometer which is an active microwave remote sensing instrument. ASCAT measures space-borne variables which are used in numerical weather prediction and tropical cyclone forecasts [16]. In 2008, the European Organization for the Exploitation of Meteorological Satellites (EUMETSAT) developed an ASCAT soil moisture processing and dissemination service in cooperation with Vienna University of Technology. For this study, EUMETSAT H113 and H114 products were obtained as saturation index values. The data were retrieved from the website (hsaf.meteoam.it) of EUMETSAT Satellite Application Facility to Support the Operational Hydrology and Water Management (H-SAF). For comparison purposes with the other soil moisture products, these data were converted to volumetric soil moisture content. In order to convert the saturation index to volumetric soil moisture ratio, saturation index values were multiplied by the average porosity of soil samples ($\rho_{avg} = 0.51$) obtained from the calibration studies of the CRNP located at Çakıt Basin. Porosity values for COSMOS stations were taken as the average values of porosity data obtained fromthe harmonized world soil database and GLDAS.

### 2.1.2. Soil Moisture and Ocean Salinity (SMOS)

SMOS refers to Soil Moisture and Ocean Salinity which is an Earth Observation satellite mission of the European Space Agency (ESA). SMOS was launched in November 2009 [17]. SMOS is a passive microwave remote sensing instrument, which uses Microwave Imaging Radiometer using Aperture Synthesis (MIRAS) to pick up faint microwave emissions from Earth's surface. SMOS is capable of mapping land soil moisture and ocean salinity [18]. In this study, level 3 SMOS data obtained from Barcelona Experts Center (BEC) were used for daily volumetric soil moisture data. Soil moisture data of SMOS were taken for both ascending and descending node products as well as

their average values. SMOS Level 3 soil moisture data were produced and disseminated by the BEC (www.smosbec.icm.csic.es), which is a joint initiative of the Spanish Research Council (CSIC) and the Technical University of Catalonia (UPC), mainly funded by the Spanish National Program on Space.

### 2.1.3. Soil Moisture Active and Passive (SMAP)

The Soil Moisture Active and Passive (SMAP) measures the amount of water in the top 5 cm of soil anywhere on the Earth's surface. Root zone soil moisture values have also been provided by SMAP Level 4 (L4) products. This product is also able to distinguish frozen or thawed ground [19,20]. For this study, SMAP L4 9 km EASE-Grid Surface and Root Zone Soil Moisture Analysis products were utilized. The data were retrieved from NASA's Earth Observing System Data and Information System (EOSDIS) website (earthdata.nasa.gov). In Level 4 products of SMAP geophysical parameters were derived by assimilating level 1, level 2 and level 3 products into a land surface model [21]. The temporal resolution of the product is 3 h, in this study, in order to have reasonable comparisons with the other products, daily averages of AM products (descending orbit) and PM products (ascending orbit) were utilized. According to Montzka et al. [14], SMAP provides higher accuracy soil moisture product with low noise or uncertainties as compared to other satellite soil moisture products.

### 2.1.4. Advanced Microwave Scanning Radiometer 2 (AMSR2)

AMSR2 is mainly based on its predecessor AMSR-E. AMRS2 is operated by the Japan Aerospace Exploration Agency (JAXA). AMSR2 soil moisture products are produced by using Land Parameter Retrieval Model (LPRM) which converts space-borne observed brightness and temperatures to soil moisture. There are two bands (C-band and X-band) available for AMSR2 soil moisture products [22]. In this study both descending and ascending C-band (6.93 GHz) soil moisture products of AMSR2 were used. The AMSR2 data were retrieved from the website of EOSDIS.

### 2.1.5. Climate Change Initiative (CCI)

The CCI soil moisture dataset was produced by combining different satellite soil moisture products [23–25]. In order to test the effectiveness of a combined soil moisture dataset, CCI soil moisture datasets were retrieved for active, passive and combined products from the website of ESA's Climate Change Initiative (www.esa-soilmoisture-cci.org).

### 2.1.6. Global Land Data Assimilation System (GLDAS)

GLDAS [26] provides land surface model data which are derived from a global meteorological dataset [27]. There are different land surface model outputs available within the context of GLDAS, in this study Noah LSM level 4 with 3 h temporal resolution data [28] were used. The GLDAS data were also retrieved from the website of EOSDIS. The temporal resolution was upscaled from 3 h to 1d by taking daily averages. Satellite products provide information about the soil moisture at first few centimeters of the soil whereas GLDAS and standalone Noah Land Surface Model provides soil moisture information at various depths. In this study, in order to have reasonable comparisons with the other products, soil moisture values obtained for the first soil layer (0–10 cm) were used.

### 2.1.7. Noah Land Surface Model (Noah LSM)

In this study, besides GLDAS (Section 2.1.6), a stand-alone, uncoupled and 1-D column version of Noah LSM was utilized to simulate surface soil moisture via land-surface model which makes use of the in-situ meteorological data using surface energy balance and water balance equations [29]. As a result of the advances in 2003, the model is able to represent soil moisture in four layers by calculating the movement of water between soil layers and the surface as well as the loss of water by transpiration from the upper three layers [30]. The forcing variables of Noah LSM for this study are precipitation rate, wind speed, air temperature, relative humidity, surface pressure, incoming shortwave radiation

and incoming longwave radiation which have 30 min of temporal resolution. The model was run for two water years (from October 2016 to October 2018) and soil moisture values were computed for four different layers (0.1 m, 0.3 m, 0.6 m and 1.0 m). Soil moisture values between 0–10 cm depths were used in this study in order to compare the data with satellite based soil moisture products. The input of NOAH LSM used in the analyses are summarized in Figure 1. CRNP soil moisture values were compared with a standalone Noah Land Surface Model which was run by using in-situ meteorological data for the site in Turkey.

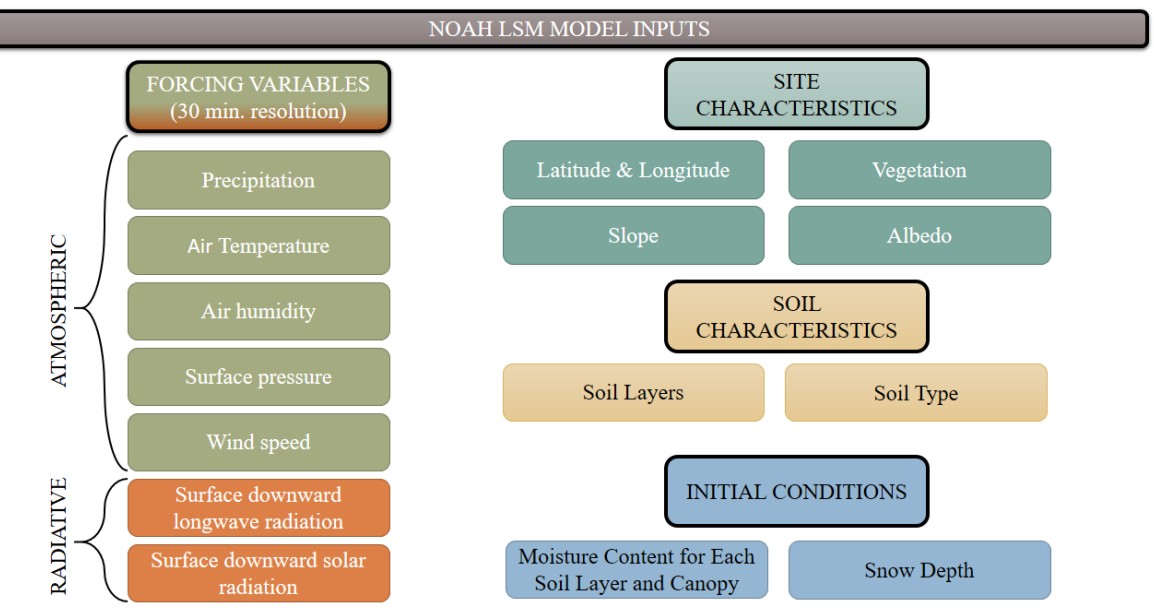

**Figure 1.** Input Parameters of NOAH Land Surface Model those are used in the analyses.

### 2.2. Cosmic Ray Neutron Probe Data

#### 2.2.1. COSMOS Database

The COsmic-ray Soil Moisture Observing System (COSMOS) has more than one hundred CRNP stations with available data on the Internet which are accessible via the project website (cosmos.hwr.arizona.edu). CRNP stations of COSMOS database count neutrons to infer soil moisture. The data are being used to understand the hydrological processes and calibration/validation of remote sensing products [31]. Data of all 104 stations, which are available in the COSMOS database, were investigated in this study.

#### 2.2.2. CRNP at Çakıt Basin

The study area (Çakıt Basin) is located at the south part of Turkey, having 526 km$^2$ area and the elevation of the basin varies from 963 m to 3450 m, where the mean elevation is 1600 m (Figure 2a). The natural vegetation is composed of pasture and shrub, there are also newly planted cherry trees. A CRS200B soil moisture probe (Hydroinnova, Albuquerque, NM, USA) was installed at elevation of 1459 m (37.51548° N 34.497877° E) (Figure 2b). Soil sampling was conducted between December 3 to 5, 2016 within the theoretical footprint of the CRNP (a circle centering the CRNP with 670 m diameter [5]). According to [31] the calibration of a CRNP at sea level should be conducted with a total number of 108 soil samples at 18 different locations for 6 layers of soil (5-cm increments from 0 to 30 cm). The distance between the sampling locations and the CRNP should be 25, 75, and 175 m. Samples were taken from 18 different locations in accordance with [31] and within the footprint of CRNP (Figure 2c).

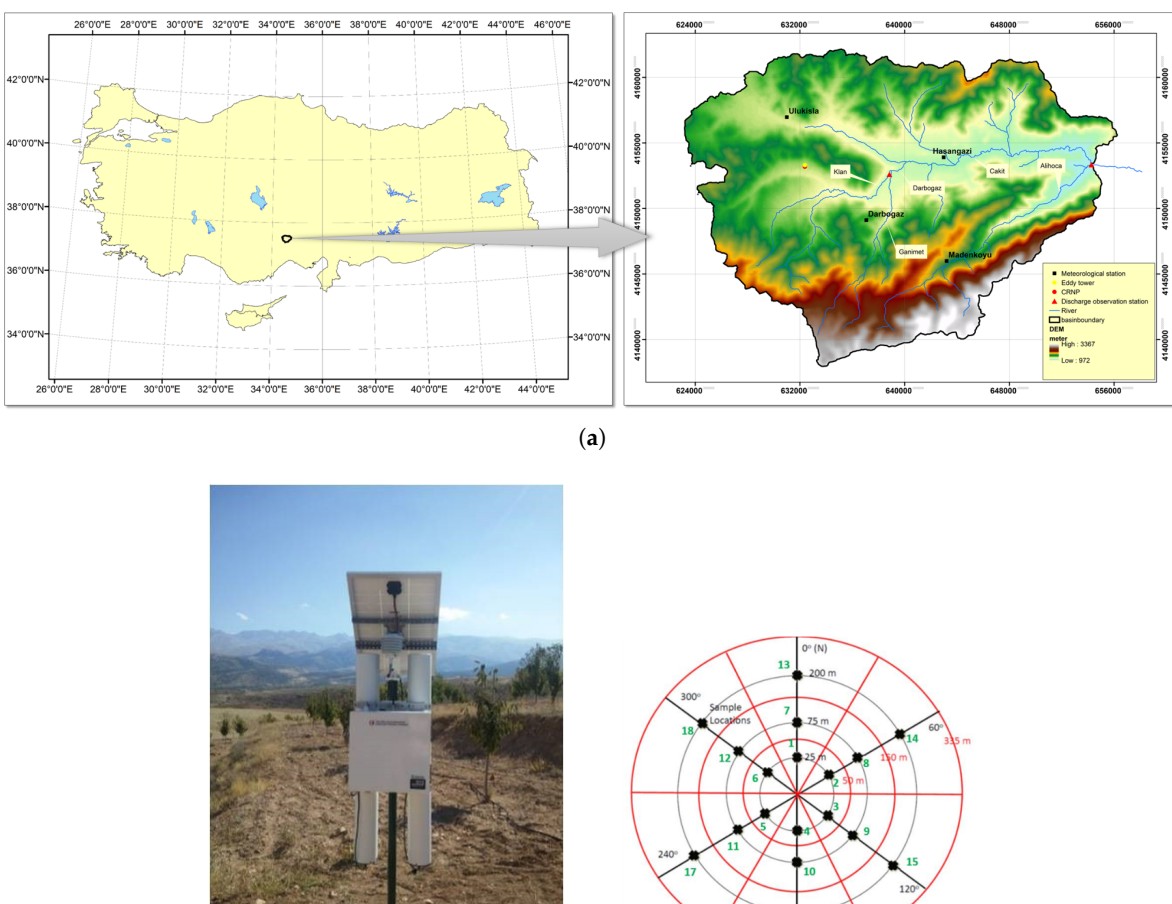

**Figure 2.** Cosmic Ray Neutron Probe Installed at Çakıt Basin-Turkey (**a**) Location of the Çakıt Basin (**b**) Cosmic Ray Neutron Probe (**c**) Soil Sampling Locations. (Black: Soil sample location distances from the center in meter, Green: Sample numbers)

The results of the calibration study show that; mean bulk density is 1.495 g/cm$^3$, gravimetric soil moisture is 0.148 kg/kg, and the reference neutron counting rate is $N_0$: 1933.4 cph. The CRNP was installed in Çakıt Basin on 6 October 2016. The sensor has been providing real-time hourly data since 11 November 2016. The data between 30 December 2016 and 3 January 2017 is missing due to a technical problem caused by excessive snow accumulation on the sensor (Figure 3b). For this study, soil moisture data obtained by a TDR sensor installed at 5 cm depth in the ground within the foot print of CRNP and the meteorological parameters obtained from a meteorological observation station installed in the close vicinity to the study area were used. Neutron counts (CRNP), precipitation, mean temperature, soil moisture measurements (TDR), soil temperature and snow depth data for the observation period are provided in Figure 3a.

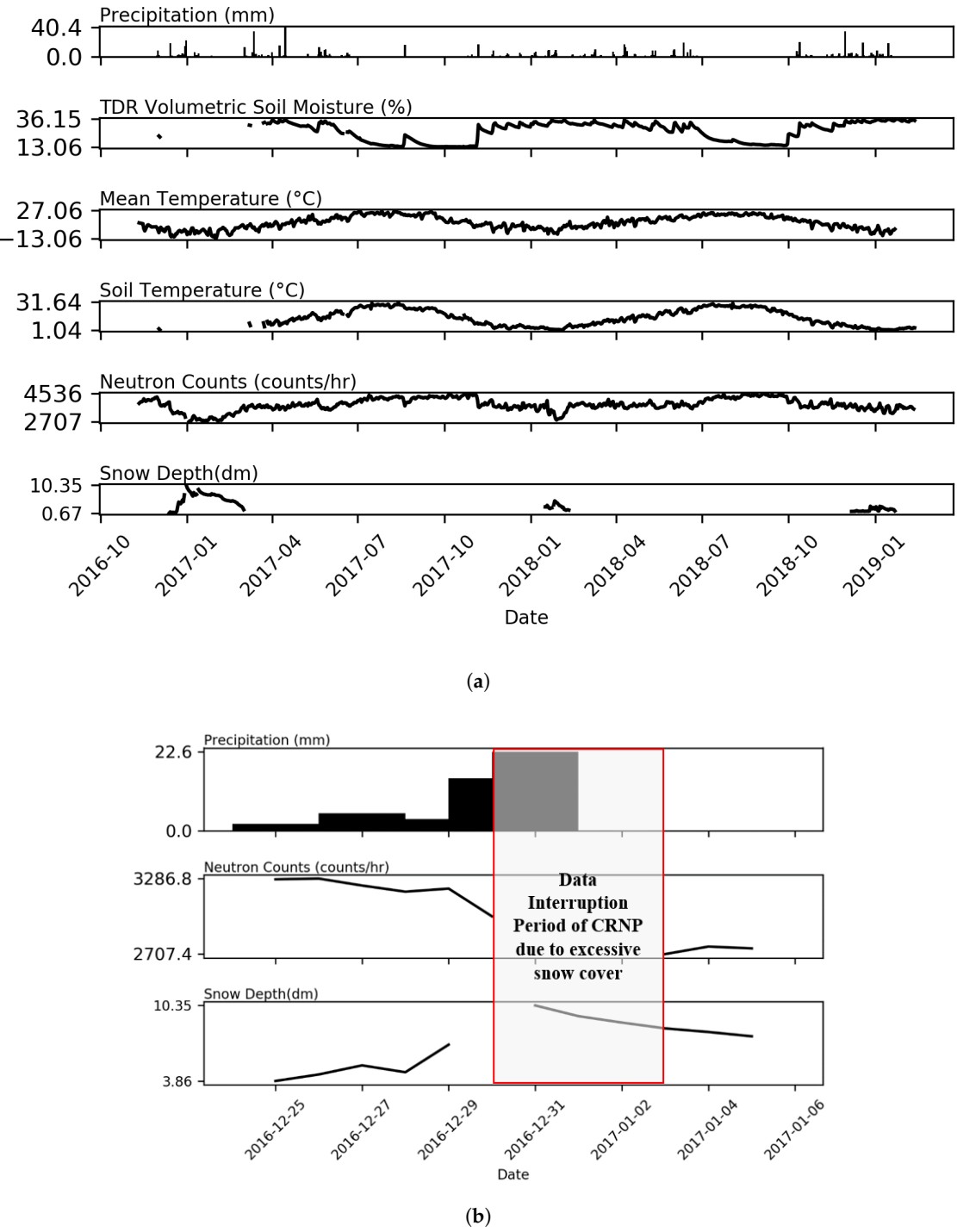

**Figure 3.** (**a**) Precipitation, mean temperature, soil moisture measurements (TDR), soil temperature and snow height data observed at the site. (**b**) Data interruption period of Çakıt CRNP.

### 2.3. Methodology

#### 2.3.1. Linear Correlations and Root Mean Square Errors

The coefficient of determination ($r^2$) value (Equation (1)) is the direct indicator of a linear relation between two datasets. High $r^2$ value between two different soil moisture datasets indicates similar responses to storm events. However, $r^2$ value is not affected by systematic biases. For this reason, root

mean square error (RMSE) (Equation (2)) and unbiased root mean square error (ubRMSE) (Equation (3)) values were also investigated for comparisons between CRNPs and satellite soil moisture products.

$$r^2 = \frac{N \sum xy - \sum x \sum y}{\sqrt{\left[ N \sum x^2 - (\sum x)^2 \right] \left[ N \sum y^2 - (\sum y)^2 \right]}} \tag{1}$$

where $x$ and $y$ are the two datasets and $N$ is the number of data points.

$$\text{RMSE} = \sqrt{\frac{1}{N} \sum_{t=1}^{N} e_t^2} \tag{2}$$

where $e_t$ is the difference between two data points and $n$ is the $x$ and $y$ are the two datasets and $N$ is the number of data points.

$$ubRMSE = \sqrt{\frac{\sum_{i=1}^{n} \left[ (G_i - \overline{G}) - (S_i - \overline{S}) \right]^2}{n}} \tag{3}$$

where $G_i$ is the ground observation data, $\overline{G}$ is the mean of the ground observations data, $S_i$ is the satellite observation data, $\overline{S}$ is the mean of the satellite observations data.

### 2.3.2. Triple Collocation

The triple collocation method was also utilized on soil moisture data sets in order to distinguish uncertainties in different types of datasets [32]. Three different types of datasets were selected for the triple collocation study; the first one is a ground measurement (CRNP), the second one is a land surface model output (GLDAS) and the third one is one of the satellite based soil moisture products (SMAP surface or rootzone products). SMAP products were selected to represent satellite based soil moisture products since they have more continuous data which is essential for triple collocation. In total there are two different triplets of data that were used in the triple collocation study for each station in the COSMOS database.

## 3. Analyses and Discussion

### 3.1. Obtaining Soil Moisture from CRNP Neutron Counts in Çakıt Basin

CRNP is able to count the neutrons which have been moderated by hydrogen atoms existing in the soil water. There is an inverse relation between the number of moderated neutrons and soil moisture intensity. The relationship between the soil moisture content and the neutron counts from CRNP have been defined in Equation (4) [33].

$$\theta(N) = \frac{a_0}{\left( \frac{N}{N_0} \right) - a_1} - a_2 \tag{4}$$

where $\theta$ is the volumetric water content, $N$ is the Neutron counting rate normalized to reference pressure and neutron intensity, $N_0$ is the neutron counting rate of dry soil under the same reference conditions, $a_0(0.0808)$, $a_1(0.372)$ and $a_2(0.115)$ are the constants of the calibration. Equation (4) was modified to account for the amount of hydrogen available within lattice water [34] and soil organic matter [35] as given in Equation (5).

$$\theta(N) = \left( \frac{a_0}{\left( \frac{N}{N_0} \right) - a_1} - a_2 - w_{lat} - w_{som} \right) \rho_{bd} \tag{5}$$

where $\rho_{bd}$ is the bulk density of soil (g/cm$^3$), $w_{lat}$ is lattice water and $w_{som}$ is the water equivalent of soil organic matter.

In order to use the neutron data obtained from CRNP for calculating soil moisture content, the measured neutron counts have to be corrected by considering environmental factors. Each CRNP measurement will reflect its own site characteristics for elevation, air pressure and absolute humidity. However, Equations (4) and (5) have been developed for a reference pressure condition. Additionally, intensity of the incoming neutron flux also effects the number of neutrons counted. Hence, in order to convert CRNP neutron counts to soil moisture content, before utilizing Equations (4) and (5), neutron counts have to be corrected for atmospheric pressure variation by using Equation (6) [36], for atmospheric water vapor by using Equation (7) [37] and for the intensity of the incoming neutron flux by using Equation (8) [31].

$$f_{bar} = e^{\beta*(P-P_{ref})} \tag{6}$$

where, $f_{bar}$ is the correction factor for Atmospheric Pressure Variation, $P$ is the Atmospheric pressure $P_{ref}$ is the reference atmospheric pressure at sea level (1013.25 hPa) and $\beta$ is the atmospheric attenuation coefficient (cm$^2$ g$^{-1}$ or m b$^{-1}$).

$$f_{hum} = 1 + 0.0054(p_{v0} - p_{v0}^{ref}) \tag{7}$$

where $f_{hum}$ is the correction factor for changes in atmospheric water vapor, $p_{v0}^{ref}$ is the reference absolute humidity (g m$^{-3}$) and $p_{v0}$ is the Near-surface absolute humidity (g m$^{-3}$).

$$f_{int} = \frac{I_{ref}}{I_m} \tag{8}$$

where $f_{int}$ is the correction factor for incoming neutron intensity, $I_{ref}$ is the reference counting rate for the same neutron monitor from an arbitrary fixed point in time and $I_m$ is the selected neutron monitor counting rate at any desired point in time.

Soil water content does not affect the high-energy secondary neutron flux, thus it is possible to use high-energy secondary neutron flux measurements in the neutron flux intensity correction. There are neutron monitors measuring the flux of high-energy secondary neutrons around the globe [38]. The COSMOS network in USA and the CosmOz network in Australia take the date 1 May 2011 as the fixed point in time for $I_{ref}$. Any other arbitrary fixed point in time may also be used for intensity corrections. Real-time neutron intensity data for various neutron monitoring stations around the globe is available in the Neutron Monitor Database (NMDB; www.nmdb.eu) founded under the European Union's FP7 programme (contact no. 213007) [35]. In this study, the Athens NMDB station was used for intensity correction since it has a similar geomagnetic cutoff rigidity (8 GV) with Çakıt Basin. Corrected neutron flux was calculated using Equation (9). [35]

$$N = (f_{bar} f_{hum} f_{int}) N_{raw} \tag{9}$$

where $N$ is the corrected neutron flux, $N_{raw}$ is the uncorrected neutron count from the CRNP.

Variables used for environmental correction and calculated environmental correction factors for the CRNP located at Çakıt Basin are shown in Figure 4.

CRNPs can provide soil moisture information for several hectares due to its intermediate horizontal footprint [5] which makes the validation of satellite soil moisture products with CRNPs reasonable. However, the vertical footprint of CRNP is not constant and the effective measurement depth can be calculated as a function of soil moisture content [39]. Considering the vertical footprint, especially for wetter conditions, Root Zone Soil Moisture (RZSM) cannot be directly estimated by CRNPs but CRNP data can be used together with ancillary soil moisture datasets to estimate RZSM values. For higher soil moisture values with less effective depth, contribution of CRNPs in determination of RZSM decreases [40]. It should also be noted that the above ground biomass and

canopy interception have strong influence on cosmic ray neutron intensity [41] which means soil moisture inferring methods which do not take biomass into account may yield inaccurate results.

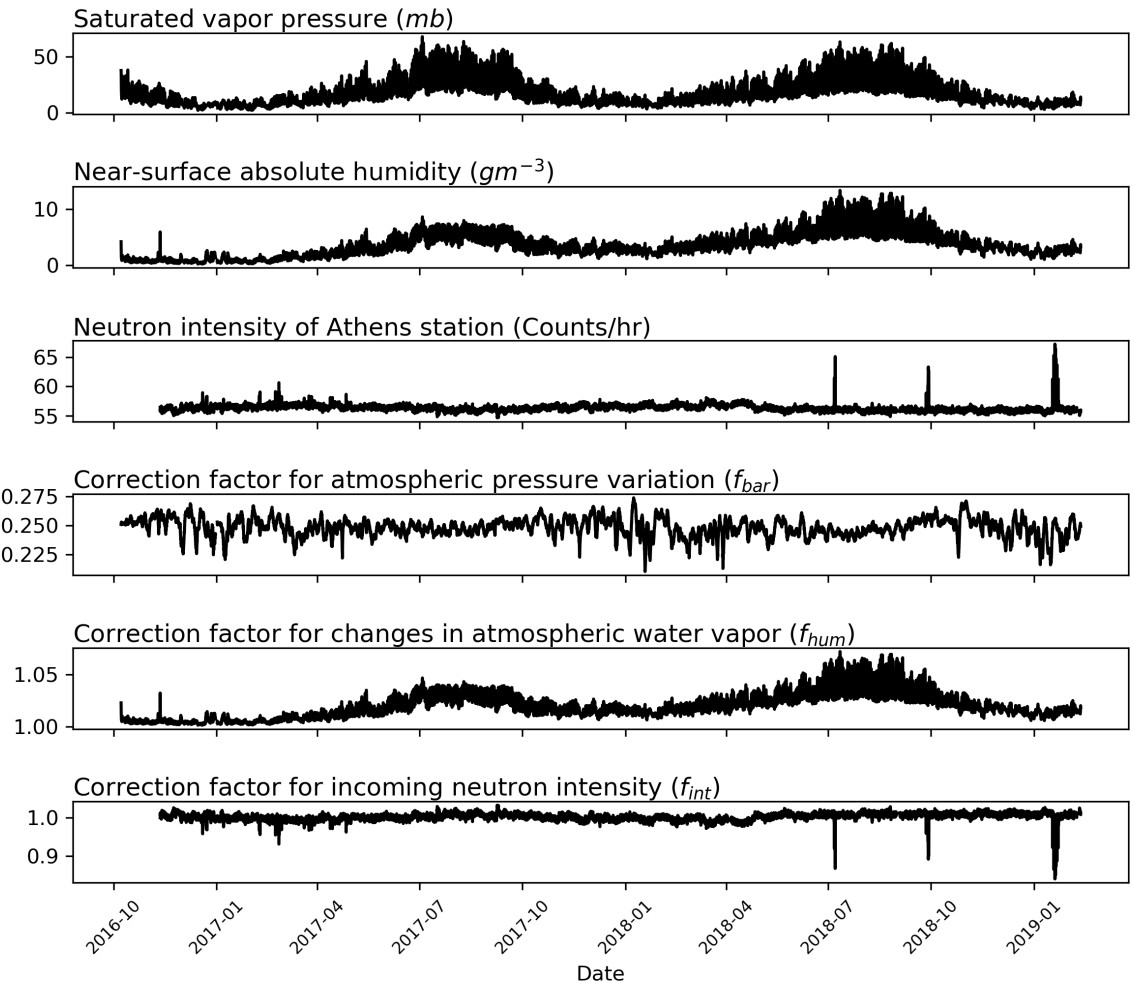

**Figure 4.** Variables Used for Environmental Correction and Environmental Correction Factors.

### 3.2. Validation with COSMOS Database

Among all spaceborne products, SMAP has the shortest range of time since its data starts from March 2015 and the other products start earlier. In order to have consistent comparisons, timescale of the analyses was taken as March 2015 to December 2018. The data from all 104 stations, which are available in the COSMOS database, were investigated in this study and the data from 82 stations were available after March 2015, thus the remaining 22 stations could not be used in the analyses. Comparisons between space-borne datasets and COSMOS stations were conducted for ascending daily products, descending daily products and daily averages of the products. Soil moisture values obtained from COSMOS Database are compared with the soil moisture values obtained from the nearest pixel of satellite products to each COSMOS station. Snow filtering was conducted by making use of the SMAP snow height data [21] and ASCAT snow probabilities. The days with SMAP snow height data is greater than 8 mm and at the same time ASCAT snow probability data is greater than 10% are taken as snowy days and excluded from the analyses. The coefficient of determination ($r^2$) and ubRMSE values are determined for each station and the satellite soil moisture values at the nearest pixel for each satellite product. Box and whisker plots that summarizes $r^2$, ubRMSE and bias values for all of the comparisons between satellite products and the CRNPs in COSMOS database are given in Figures 5–7. Triple collocation analyses were performed by making use of observed(CRNP), modeled(GLDAS) and

satellite(SMAP) soil moisture data for the stations of the COSMOS database including the Çakıt CRNP. The analyses were made for the stations that have at least 100 data points for the given three products in common. In Figure 8 the results of the triple collocation statistics are summarized.

### 3.3. Validation with CRNP in Çakıt Basin

The analyses for Çakıt Basin (Figure 2a) were conducted in the same way as it was done with the stations in the COSMOS database. ASCAT, SMOS, SMAP, AMSR2, CCI and GLDAS datasets as it is mentioned in Section 3.2, were used. In this section, the relations between different types of soil moisture values obtained from different products are investigated in more detail. In addition to these analyses of satellite soil moisture products, the soil moisture values retrieved from the CRNP located at the Çakıt Basin were also compared with TDR soil moisture values obtained from the site as well as the calculated soil moisture values of a Noah Land Surface Model that was established using the observed site conditions and in-situ meteorological data. Daily resampled time series of all soil moisture products are shown in Figure 9. There are unrealistically high values in CRNP time series which are occurring in winter days due to snow cover and they were excluded from the analyses by using exactly the same methodology which was previously described in Section 3.2.

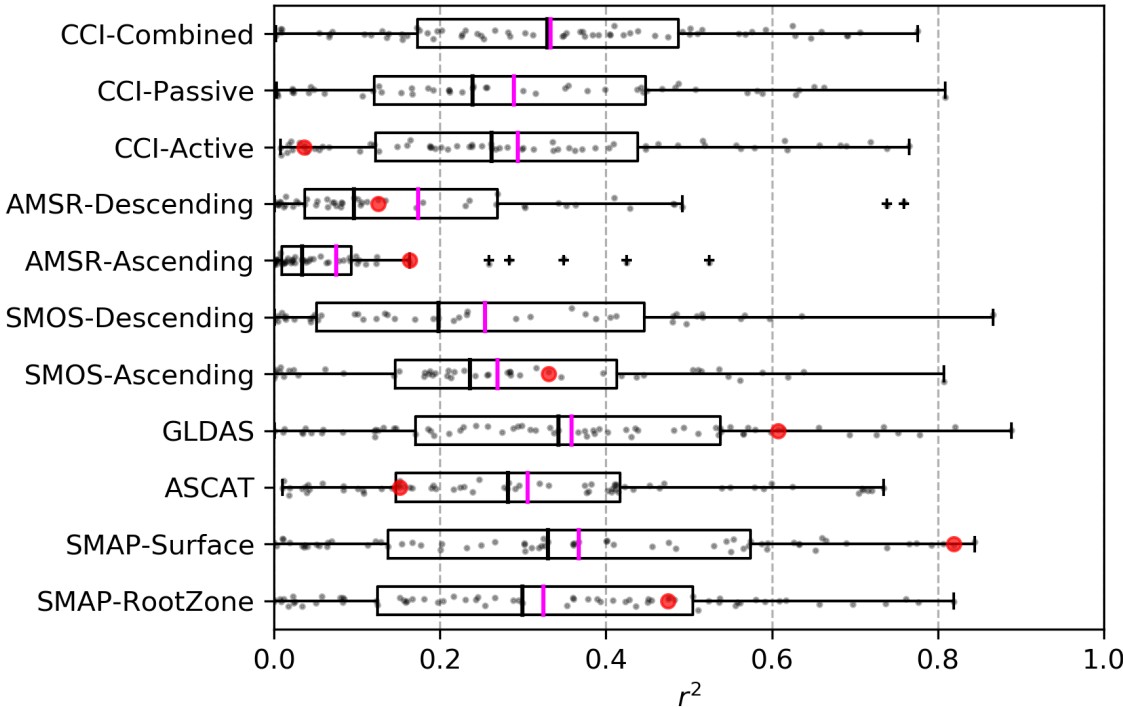

**Figure 5.** Box and whisker plots showing $r^2$ values between the satellite products and the CRNP's in the COSMOS database. Red points show the results for Çakıt Basin CRNP, pink lines show the mean values and + signs depict outliers.

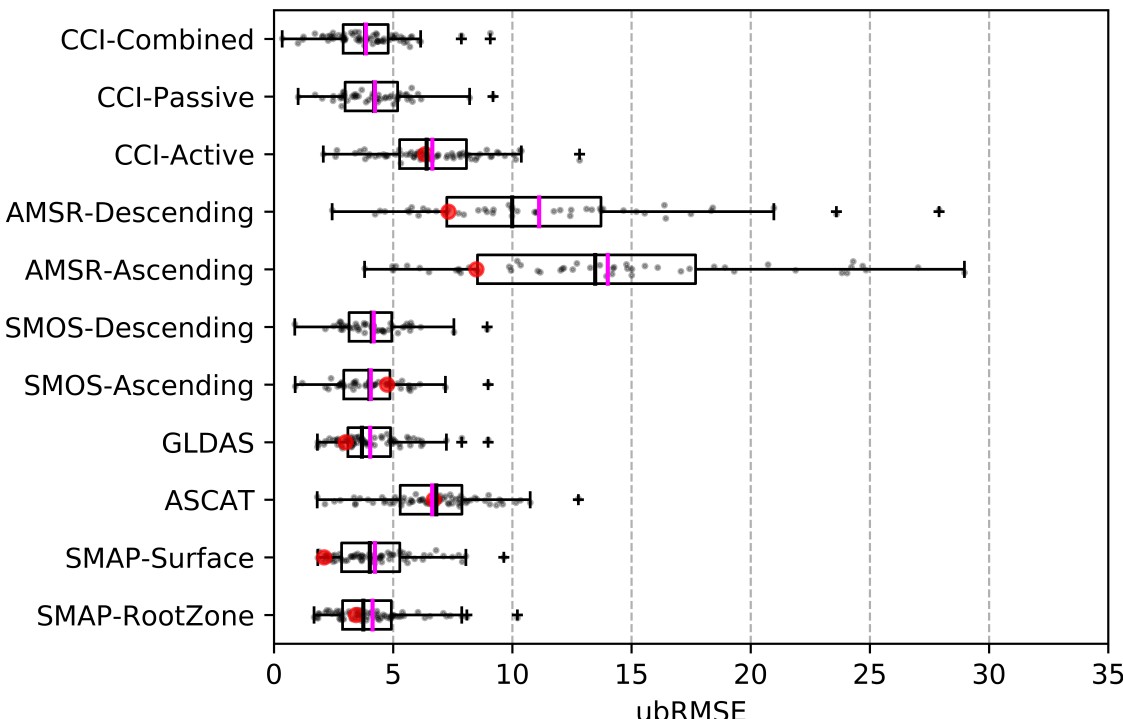

**Figure 6.** Box and whisker plots showing ubRMSE values between the satellite products and the CRNP's in the COSMOS database. Red points show the results for Çakıt Basin CRNP pink lines show the mean values and + signs depict outliers.

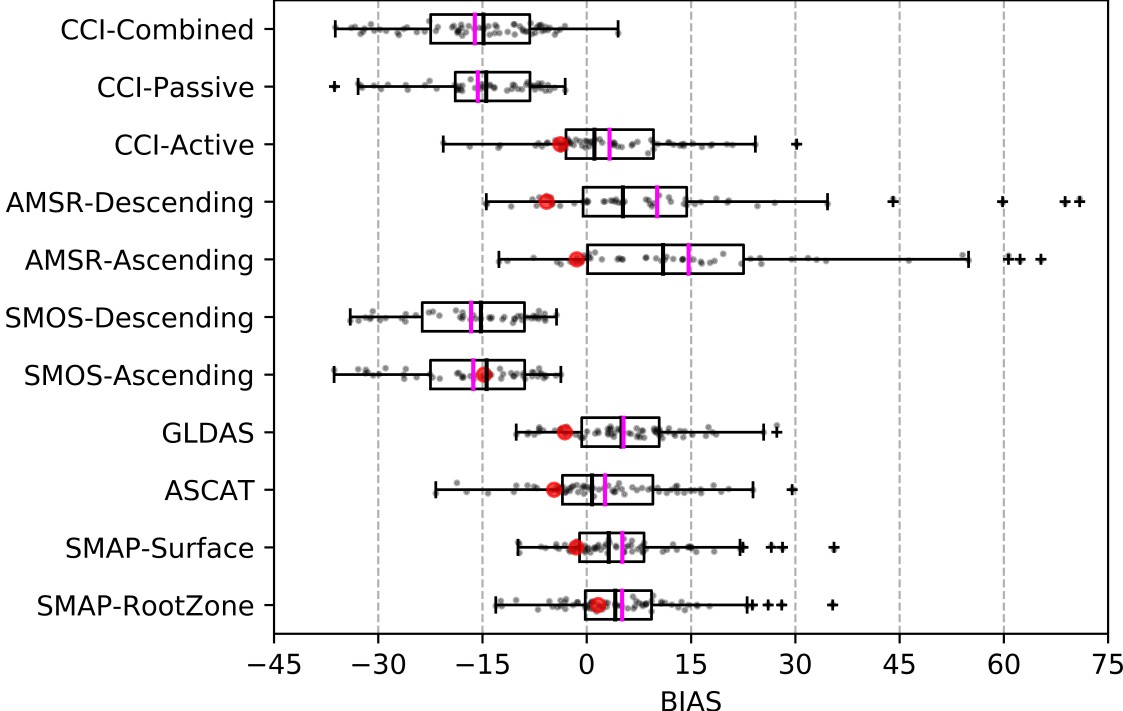

**Figure 7.** Box and whisker plots showing bias values between the satellite products and the CRNP's in the COSMOS database. Red points show the results for Çakıt Basin CRNP , pink lines show the mean values and + signs depict outliers.

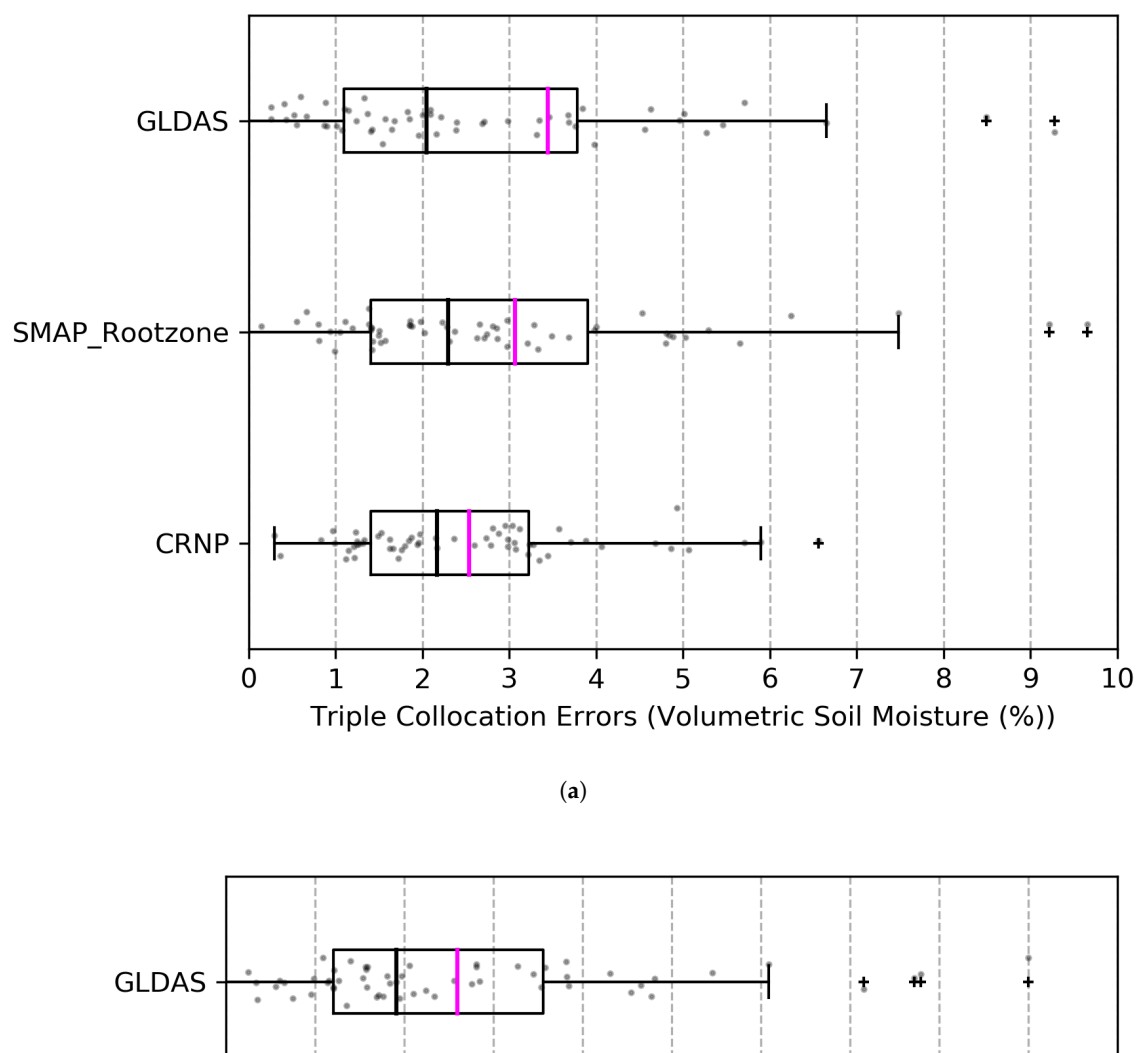

(**a**)

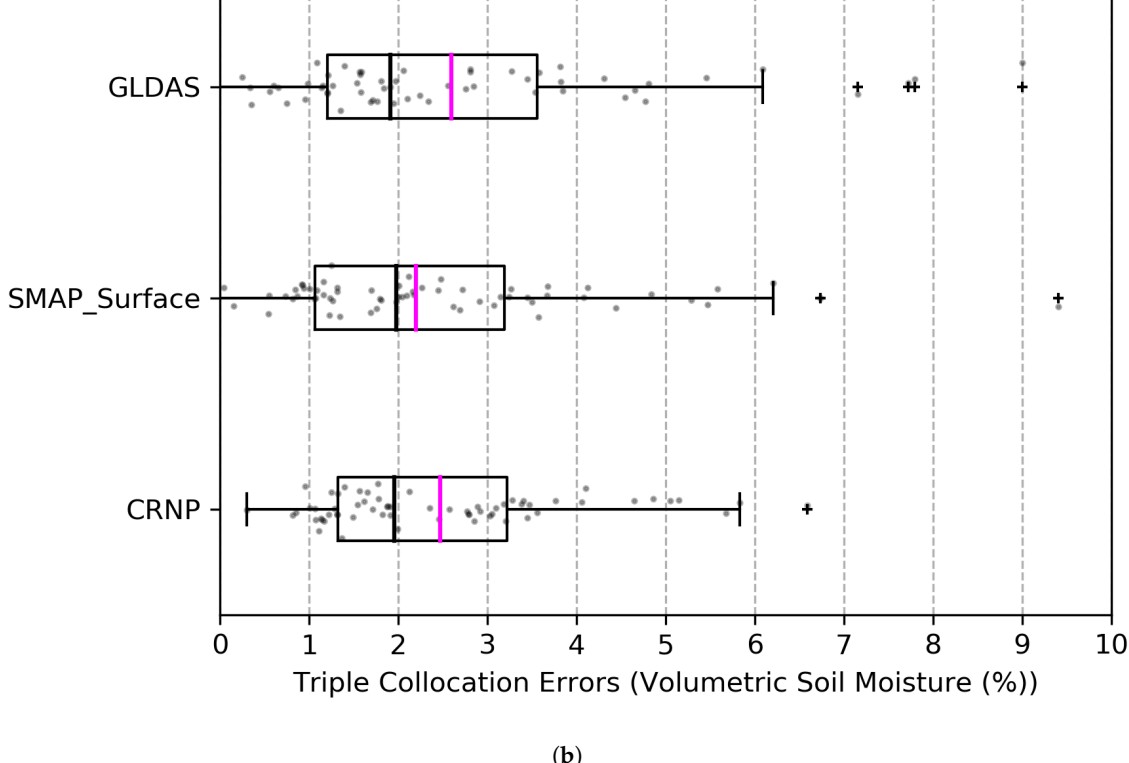

(**b**)

**Figure 8.** Triple collocation errors of soil moisture products for observed (CRNP), modelled (GLDAS) and satellite (SMAP) for COSMOS database, pink lines show the mean values and + signs depict outliers. (**a**) Triple collocation errors of GLDAS–SMAP Rootzone–CRNP. (**b**) Triple collocation errors of GLDAS-SMAP Surface-CRNP.

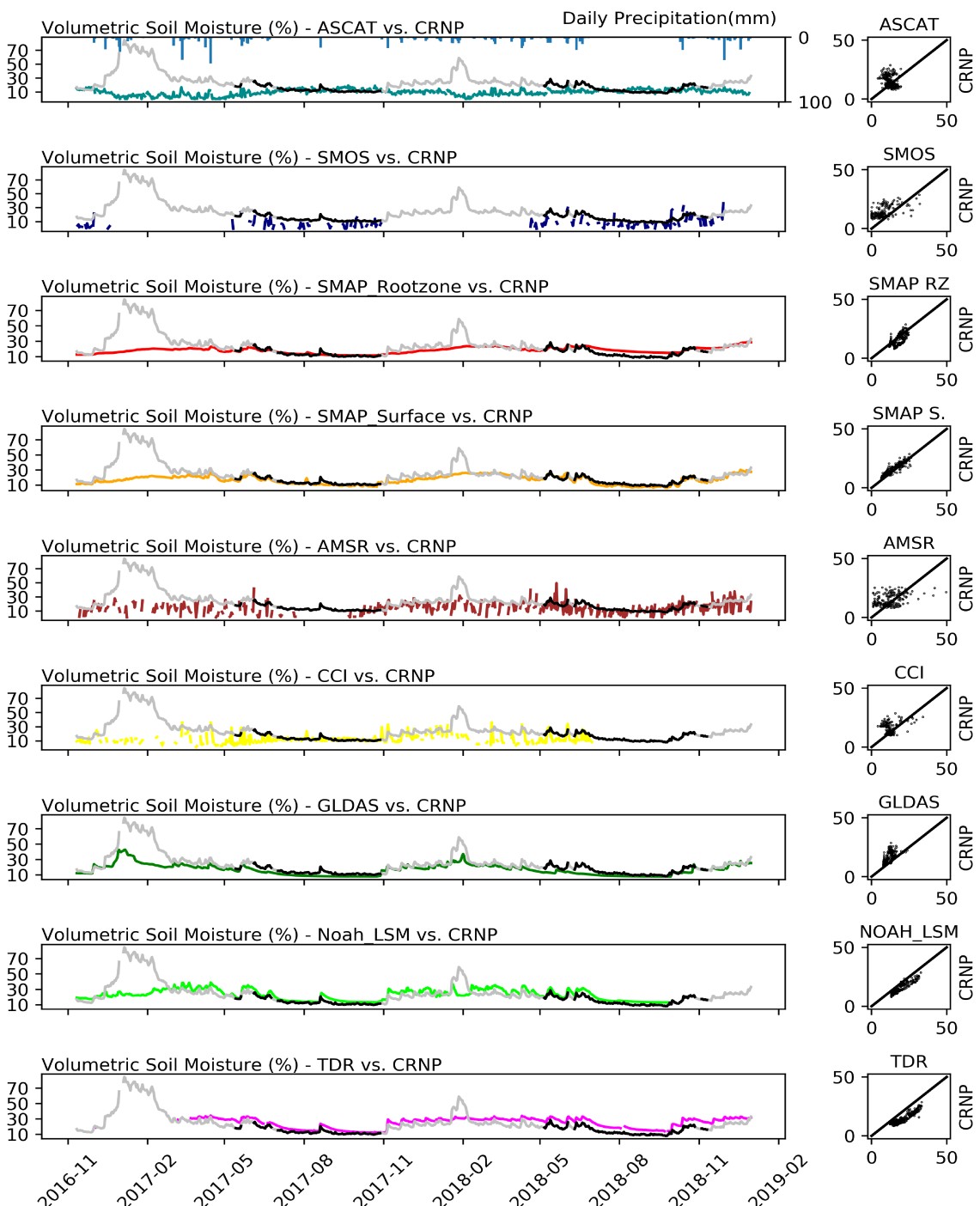

**Figure 9.** Time series of satellite soil moisture products. Filtered days due to snow are shown in grey, CRNP soil moisture values are shown in black, other soil moisture products are shown in different colors.

*3.4. Discussions*

3.4.1. COSMOS Database Results

Among the satellite soil moisture products validated with the COSMOS database, ASCAT, CCI and SMAP products' results are better compared to others. The mean $r^2$ value for the ASCAT product is around 0.32, whereas the mean Root Mean Square Error (RMSE) is around 0.11 $m^3/m^3$ and unbiased

Root Mean Square Error (ubRMSE) is 0.07 m$^3$/m$^3$. These results are in good agreement with the results of [15] in which in situ measurements from the FLUXNET observational network were used in the validation of the ASCAT soil moisture product. Figures 5–7 and Table 2 show that the ascending node products of passive remote sensing products are slightly better than descending ones as it was previously underlined by [13]. The CCI combined product shoedw very good $r^2$ values with CRNP's, it is as successful as the SMAP products but not better than them, although CCI is a combination of different products including SMAP.

**Table 2.** Mean and Median Values of $r^2$, RMSE, ubRMSE and bias values of soil moisture products with CRNPs

| Soil Moisture Products | $r^2$ Mean | $r^2$ Median | RMSE Mean | RMSE Median | ubRMSE Mean | ubRMSE Median | Bias Mean | Bias Median |
|---|---|---|---|---|---|---|---|---|
| SMAP-RootZone | 0.32 | 0.30 | 9.37 | 7.08 | 4.13 | 3.75 | 5.09 | 4.09 |
| SMAP-Surface | 0.37 | 0.33 | 9.06 | 6.95 | 4.24 | 4.02 | 5.10 | 3.17 |
| ASCAT | 0.31 | 0.28 | 10.94 | 9.05 | 6.64 | 6.81 | 2.61 | 0.76 |
| GLDAS | 0.36 | 0.34 | 9.23 | 8.40 | 4.04 | 3.69 | 5.27 | 5.00 |
| SMOS-Ascending | 0.27 | 0.24 | 16.98 | 15.34 | 4.06 | 4.01 | −16.34 | −14.40 |
| SMOS-Descending | 0.25 | 0.20 | 17.31 | 16.10 | 4.19 | 4.10 | −16.64 | −15.24 |
| AMSR2-Ascending | 0.07 | 0.03 | 23.64 | 17.33 | 14.00 | 13.47 | 14.64 | 10.96 |
| AMSR2-Descending | 0.17 | 0.10 | 19.18 | 14.33 | 11.12 | 9.99 | 10.12 | 5.19 |
| CCI-Active | 0.29 | 0.26 | 10.83 | 8.87 | 6.64 | 6.41 | 3.27 | 1.10 |
| CCI-Passive | 0.29 | 0.24 | 16.38 | 15.23 | 4.23 | 4.20 | −15.67 | −14.47 |
| CCI-Combined | 0.33 | 0.33 | 16.83 | 15.80 | 3.85 | 3.82 | −16.10 | −14.87 |

The $r^2$ and ubRMSE results of the ASCAT and SMAP products are given in Figures 10–13. As can be seen in Figure 12, locations with semi-arid climatic conditions tend to have greater correlation than vegetative areas for SMAP surface soil moisture values whereas Figures 10 and 11 suggests that ASCAT is more successful in vegetative areas. On the other hand, the global comparison of the satellite products (Figures 5–7) suggests that ASCAT product is as successful as the SMAP surface product in many regions. This result was also in line with the findings of recent studies [10,42]. It was also mentioned in Reference [14] that, among the other space-borne products, SMAP has greater correlation with CRNPs. Using alternative metrics in the evaluation of satellite soil moisture products has been recommended in the related literature, skill values of ubRSME (Table 2) show that all datasets except AMSR2-Ascending and AMSR2-Descending, are similar for the CRNP stations in the COSMOS database.

The bias of SMAP-RootZone, SMAP-Surface, ASCAT, GLDAS, AMSR2-Ascending, AMSR2-Descending, CCI-Active shows under and overestimations for different sites. The bias of SMOS-Descending, SMOS-Ascending and CCI-Passive shows a trend to underestimate CRNP records at most sites. This may be due to uncertainties in the soil porosity information used for scaling the soil moisture time series for ASCAT, soil moisture retrieval algorithms of the satellite products and the horizontal and vertical scale mismatch of CRNP observations.

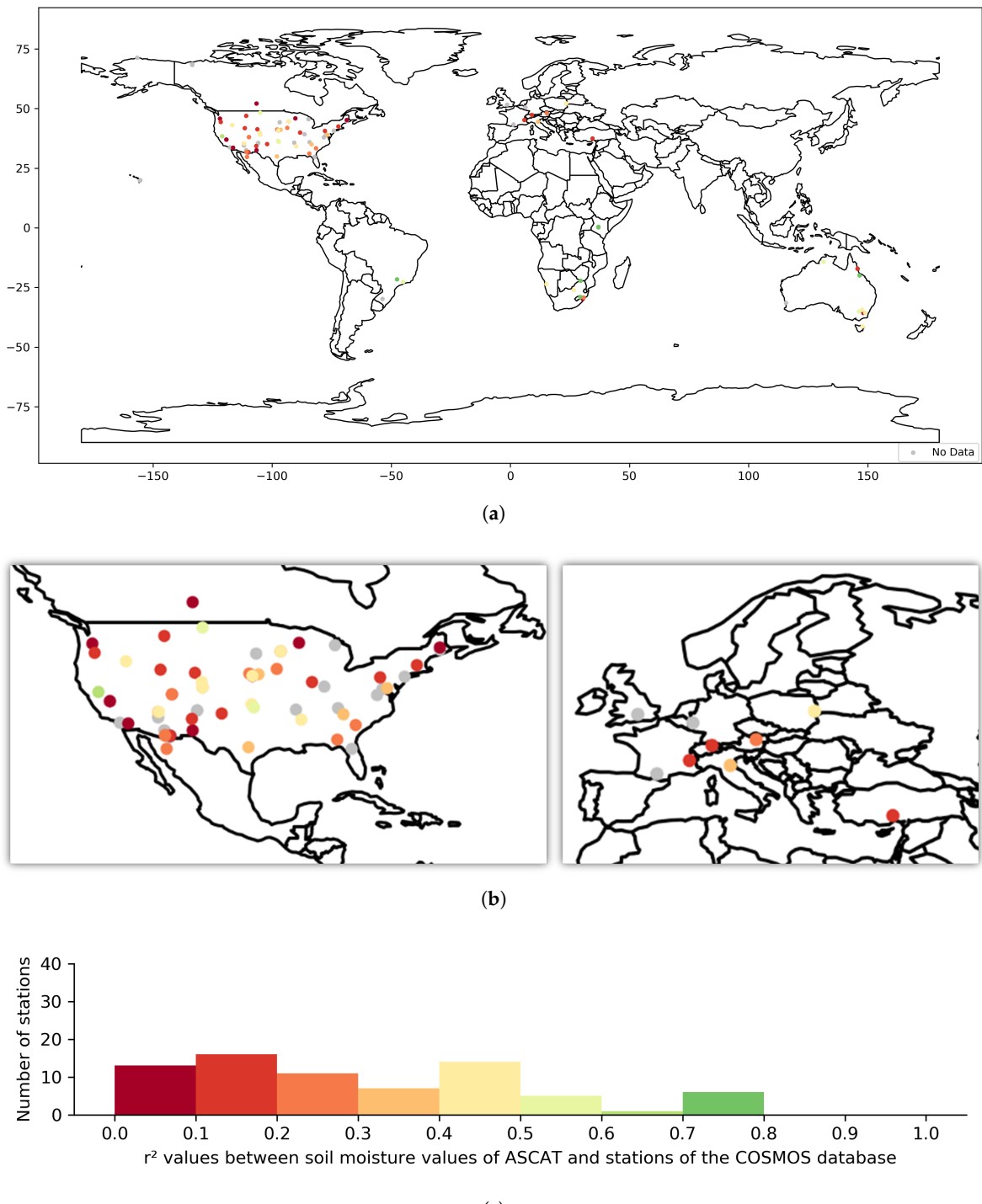

**Figure 10.** (**a**) $r^2$ values between ASCAT and COSMOS stations shown on the world map (**b**) Detailed maps of $r^2$ values for the North America and Europe (**c**) Histogram of $r^2$ values between ASCAT and COSMOS stations

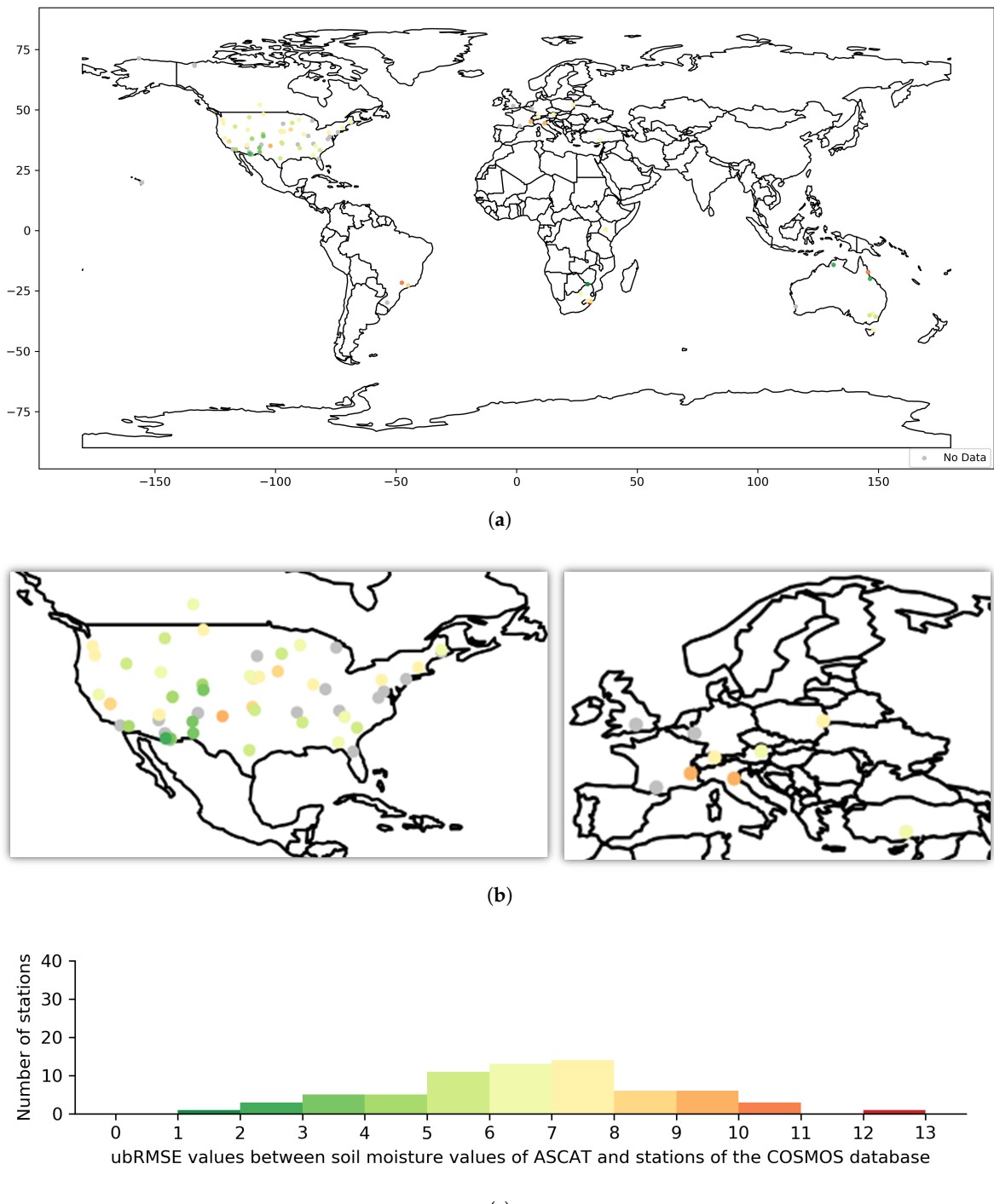

**Figure 11.** (**a**) ubRMSE values between ASCAT and COSMOS stations shown on the world map (**b**) Detailed maps of ubRMSE values for the North America and Europe (**c**) Histogram of ubRMSE values between ASCAT and COSMOS stations.

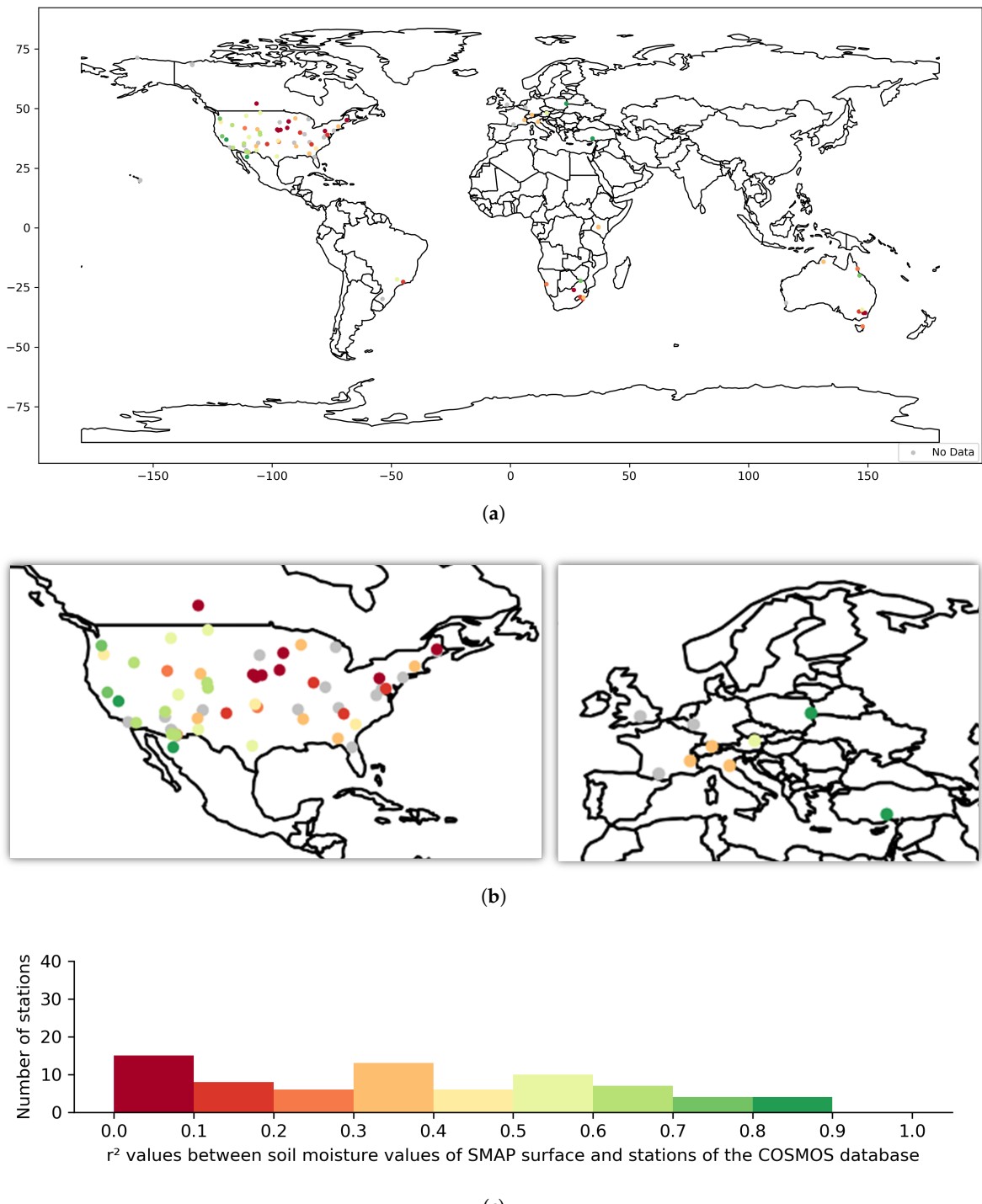

**Figure 12.** (**a**) $r^2$ values between SMAP and COSMOS stations shown on the world map (**b**) Detailed maps of $r^2$ values for the North America and Europe (**c**) Histogram of $r^2$ values between SMAP and COSMOS stations.

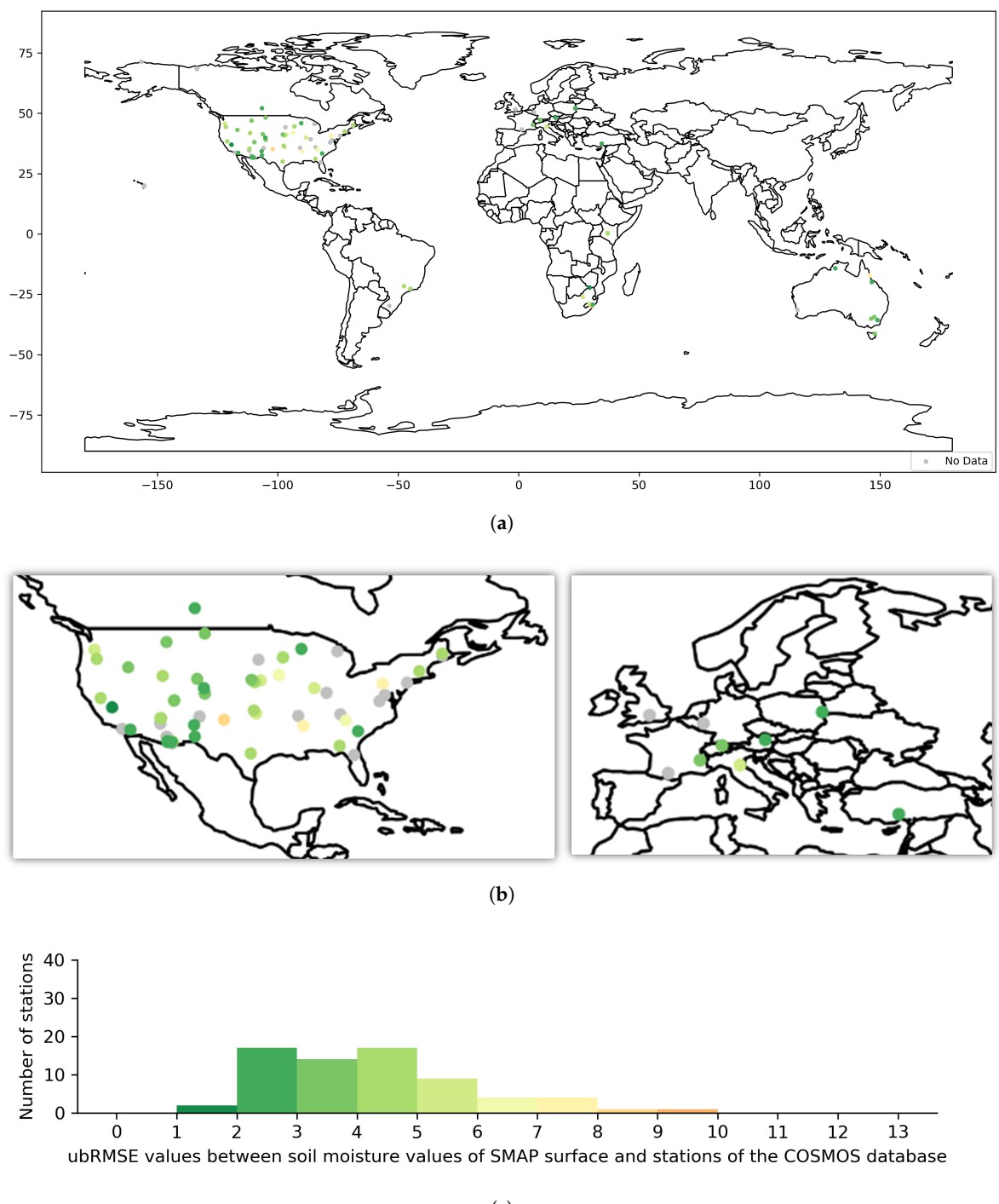

**Figure 13.** (**a**) ubRMSE values between SMAP and COSMOS stations shown on the world map (**b**) Detailed maps of ubRMSE values for the North America and Europe (**c**) Histogram of ubRMSE values between SMAP and COSMOS stations.

The triple Collocation method was applied to the datasets CRNP as ground observation, SMAP-RootZone and SMAP-Surface as satellite products and GLDAS as a model simulation. As Reference [25] lists the assumptions made to perform the TC analysis (i) Linearity between the true soil moisture signal and the observations (ii) signal and error stationarity (iii) independency between the errors and the soil moisture signal (error orthogonality) and (iv) independency between the errors of three soil moisture time series (zero-error cross-correlation), we assume they are valid for the times series obtained from CRNP, SMAP product, which gave high $r^2$ and low ubRMSE values in

the validation analyses. Summary statistics of triple collocation shows that SMAP surface product is fairly consistent with observed and modelled soil moisture products. In most of the stations the signal to noise ratio is obtained greater than 1 dB indicating that noise in the satellite product is not larger than the soil moisture signal. No scaling has been done for CRNPs where the scaling factor ($\beta$) has been taken as 1. The average scaling value for SMAP-Surface product and GLDAS model simulations are obtained as 0.88 and 0.92, respectively. The absolute TC error standard deviations are presented in Figure 8. In principle a direct comparison of TC error standard deviations from different triplets is not valid, because these errors are strictly related to the choice of the triplet and thus cannot be compared to errors derived from another triplet [43]. The range of triple collocation errors for CRNP, SMAP-Surface and GLDAS is much less compared to the ones of the triplet CRNP, SMAP-RootZone and GLDAS. This analysis must be performed in detail considering the site characteristics and their climate conditions separately.

Although CRNP is a promising device for measuring soil moisture and validating satellite products, it is difficult to determine the performance of satellite products with CRNPs in the winter months since soil moisture under snow cover cannot yet be sensed with CRNPs. For large scale projects, where accurate soil moisture information is required for several square kilometers, satellite soil moisture information can be used in conjunction with CRNPs. However, the researchers must decide on the satellite products which they will make use of. The performance of the satellite products are not the same in all around the world. In certain site conditions, some of the satellite products perform better than the others. There are many different soil moisture satellite products which can be used separately or in a combined manner. In order to test the possible use of combined products, CCI soil moisture dataset which uses different satellite soil moisture products' information has been utilized. Nevertheless, it has been shown in this study that some of the regions, including Çakıt Basin, the CCI soil moisture product has very low correlations with ground observations, which is mainly due to the performance of the selected datasets for the combination.

### 3.4.2. Çakıt CRNP Results

Results showed a generally reasonable agreement between the SMAP-Surface product and Çakıt CRNP soil moisture measurements. RMSE is obtained as 0.026 m$^3$/m$^3$, ubRMSE is 0.021 m$^3$/m$^3$ and $r^2$ is 0.82, which is the maximum value obtained among the other soil moisture products (Figure 9). For Çakıt station, the bias of all the satellite products, except SMAP-RootZone, shows a trend to underestimate CRNP records. Scatter plots showing the relation between soil moisture products with CRNP and TDR are provided in the Appendix section(Figures A1–A8).

It should be noted that the areal representativeness of CRNP is still much lower than the footprint of coarse resolution satellite soil moisture products, this spatial mismatch affects the correlations between satellite products and CRNP. Correlations between space-borne soil moisture products and in-situ observations also depend on site conditions such as soil temperature which affects the soil moisture retrieval algorithms of the products and seasonal conditions. ASCAT product for example has more correlation with CRNP and TDR for colder days. Çakıt CRNP compared results for the SMAP surface product has a good agreement for all the months and for the observed soil temperatures. According to TDR measurements, the SMAP surface product underestimates the observed soil moisture for all the months. This indicates the areal representativeness of the CRNP in monitoring the soil moisture and use of it in validating the remote sensing products. CCI performs quite well for the summer and winter months but overestimates in spring months. The inaccurate results obtained from ASCAT products for Çakıt Basin can be explained by the sub-surface scattering phenomena which can be observed during summer and causes the overestimation of soil moisture. This behavior can also be observed at the sites located in the Mediterranean region having similar characteristic with Çakıt Basin (from personal communication with Sebastian Hahn from Vienna University of Technology). As is presented by Reference [7], several factors affect ASCAT soil moisture retrieval accuracy. The main errors of satellite soil moisture products are due to topographic complexity, high vegetation density

(e.g., pluvial forests), frozen soils, snow cover, and volume scattering in dry soils. Some of these factors are common to any soil moisture product (snow cover and frozen soils) while others are more pronounced for scatterometer-based ASCAT retrievals (i.e., topography and dry soil volume scattering) with respect to competing passive microwave products. Dry soil volume scattering is the main reason in obtaining the low $r^2$ and high RMSE values for the ASCAT product for Çakıt CRNP.

Passive soil moisture products (SMOS and AMSR2) have very low correlations and high errors when compared to the CRNP and TDR observations; additionally, their temporal resolutions are very low which means it is very hard to use them in practical studies. The GLDAS product on the other hand provides consistent results with ground observations with a slight underestimation despite its lower spatial resolution and greater grid size (25 km × 25 km). Noah LSM provides slightly better results than its global scale version GLDAS as it can be expected. But the difference between Noah LSM and GLDAS is not very significant.

Comparisons between Çakıt CRNP and TDR show that both products reacted similarly to storm events (Figure 9) with a slight overestimation of TDR which is most probably due to being located at a shallower depth (5 cm) than the vertical footprint of CRNP (12–76 cm). Çakıt CRNP also has very good correlations with the land surface models. Both Local Noah LSM and GLDAS provide consistent soil moisture data with CRNP. Noah LSM forced by in situ data is slightly better than GLDAS, as one might expect.

The uncoupled version of Noah LSM which was run by using in situ meteorological data provided very similar results with the global version (GLDAS), which was run by using global data (mostly from satellite products). This indicates that the global dataset is successful at representing local conditions for Çakıt CRNP.

For Çakıt Basin SMAP-surface product has 0.027 m$^3$/m$^3$ triple collocation error whereas GLDAS product has 0.038 m$^3$/m$^3$. The signal to noise ratios for SMAP-RootZone and SMAP-Surface are obtained as −1.69 and 6.42, respectively. This indicates that noise in the satellite product is larger than soil moisture signal for SMAP-RootZone. It is obtained that the dynamic range of SMAP-RootZone is larger than that of the CRNP and GLDAS time series for Cakit station. These results are in line with the previous study [14] which suggests that SMAP produces the least triple collocation errors. (Figure 8)

## 4. Summary and Conclusions

In this study, we have inferred soil moisture using a cosmic-ray neutron probe (CRNP) and compared the findings with satellite products and a TDR sensor placed in close vicinity to CRNP in Çakıt Basin in Turkey. In addition to the Çakıt Basin CRNP, the validation of ASCAT, SMOS, SMAP, AMSR2 and GLDAS soil moisture products were conducted by using the CRNPs of the COSMOS Database which are located all around the world.

The global scale investigation on the satellite based soil moisture products and the CRNPs of the COSMOS database showed that SMAP products and CRNP have a generally higher correlation in arid-semi arid regions than vegetative ones, whereas ASCAT is not very successful at those type of areas but the correlations between ASCAT and CRNPs are generally as good as the correlations between SMAP products and CRNPs.

Among all of the stations of the COSMOS Database, the Çakıt Basin CRNP has one of the highest coefficients of determination and the lowest RMSE values with the SMAP surface soil moisture product. This is possibly due to the semi-arid site characteristics of Çakıt Basin in which CRNP can perform better. This may be also due to the fact that there are very few COSMOS stations in a several thousand kilometers distance to Çakıt Basin and the performance of both satellite products and CRNPs should be further investigated for the Mediterranean region.

Çakıt Basin CRNP station and TDR show a high correlation for soil moisture, which proves that measurements of CRNP by means of soil water content are reliable. Both sensors show consistent changes in soil moisture due to the storm events. TDR measurements are at most 30% larger than the CRNP measurements, because of being located at a shallow depth (at 5 cm in the soil) whereas

CRNP is capable of measuring soil moisture at minimum 12 cm and maximum 72 cm depth depending on the soil moisture content. SMOS, ASCAT and AMSR2 products show larger variation and noise compared to SMAP and GLDAS soil moisture products. SMAP surface soil moisture product and GLDAS has much higher correlations with CRNP than the other products, they are also capable of providing soil moisture data even in snow covered durations and CRNP has the limitation to obtain soil moisture under snow cover. For this reason, soil moisture data obtained from these products can be used in conjunction with CRNP for soil water deficit investigations and agricultural decision making processes.

In this study, the Noah Land Surface Model was also utilized to compute soil moisture by using in-situ meteorological data and it was found that the modeled soil water content well represents the CRNP and TDR soil moisture measurements. The standalone Noah LSM model was slightly better at representing site characteristics than the Noah LSM outputs obtained from GLDAS, which uses a global dataset as input. However, GLDAS soil moisture data are very close to the Noah LSM results which may indicate that the global dataset that the GLDAS is using is reliable for the Çakıt Basin.

The results of this study suggest that the Çakıt Basin SMAP surface product is better compared to the other satellite soil moisture products. In this study, SMAP root zone and surface products were investigated separately; for further studies, CRNP soil moisture values can be investigated for different soil layers using data assimilation techniques and neutron transport models to better represent SMAP root zone soil moisture values.

CRNPs show promising potential for being used in conjunction with satellite soil moisture values especially in the field of agriculture. Practical applications of CRNPs and satellite products will help better understanding of the soil moisture deficits in very large fields with a lack of a dense soil moisture observation network. In those fields, the usage of CRNPs can lead to better irrigation management and increased yield.

**Author Contributions:** Conceptualization, M.B.D. and Z.A.; Writing-original draft preparation, M.B.D.; Writing-review and editing, M.B.D. and Z.A.

**Funding:** This research was funded by TUBITAK grant number 115Y041.

**Acknowledgments:** The authors thank the teams from COSMOS, H SAF, NASA, ESA, BEC, JAXA and NMDB for making their datasets publicly available.

**Conflicts of Interest:** The authors declare no conflict of interest.

## Appendix A

Scatter plots showing the relation between soil moisture products and CRNP, TDR are provided in this section (Figures A1–A8).

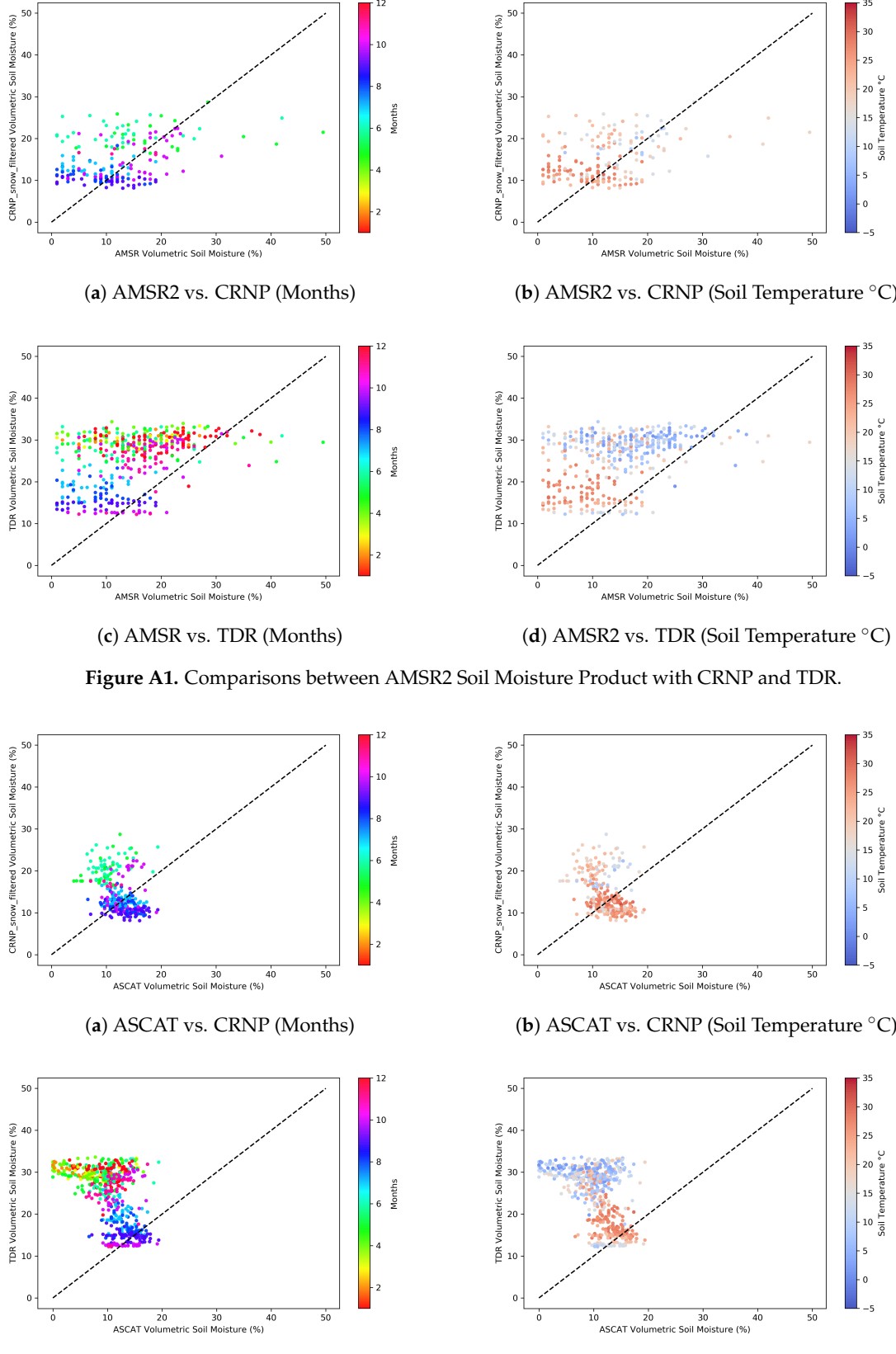

(**a**) AMSR2 vs. CRNP (Months)

(**b**) AMSR2 vs. CRNP (Soil Temperature °C)

(**c**) AMSR vs. TDR (Months)

(**d**) AMSR2 vs. TDR (Soil Temperature °C)

**Figure A1.** Comparisons between AMSR2 Soil Moisture Product with CRNP and TDR.

(**a**) ASCAT vs. CRNP (Months)

(**b**) ASCAT vs. CRNP (Soil Temperature °C)

(**c**) ASCAT vs. TDR (Months)

(**d**) ASCAT vs. TDR (Soil Temperature °C)

**Figure A2.** Comparisons between ASCAT Soil Moisture Product with CRNP and TDR.

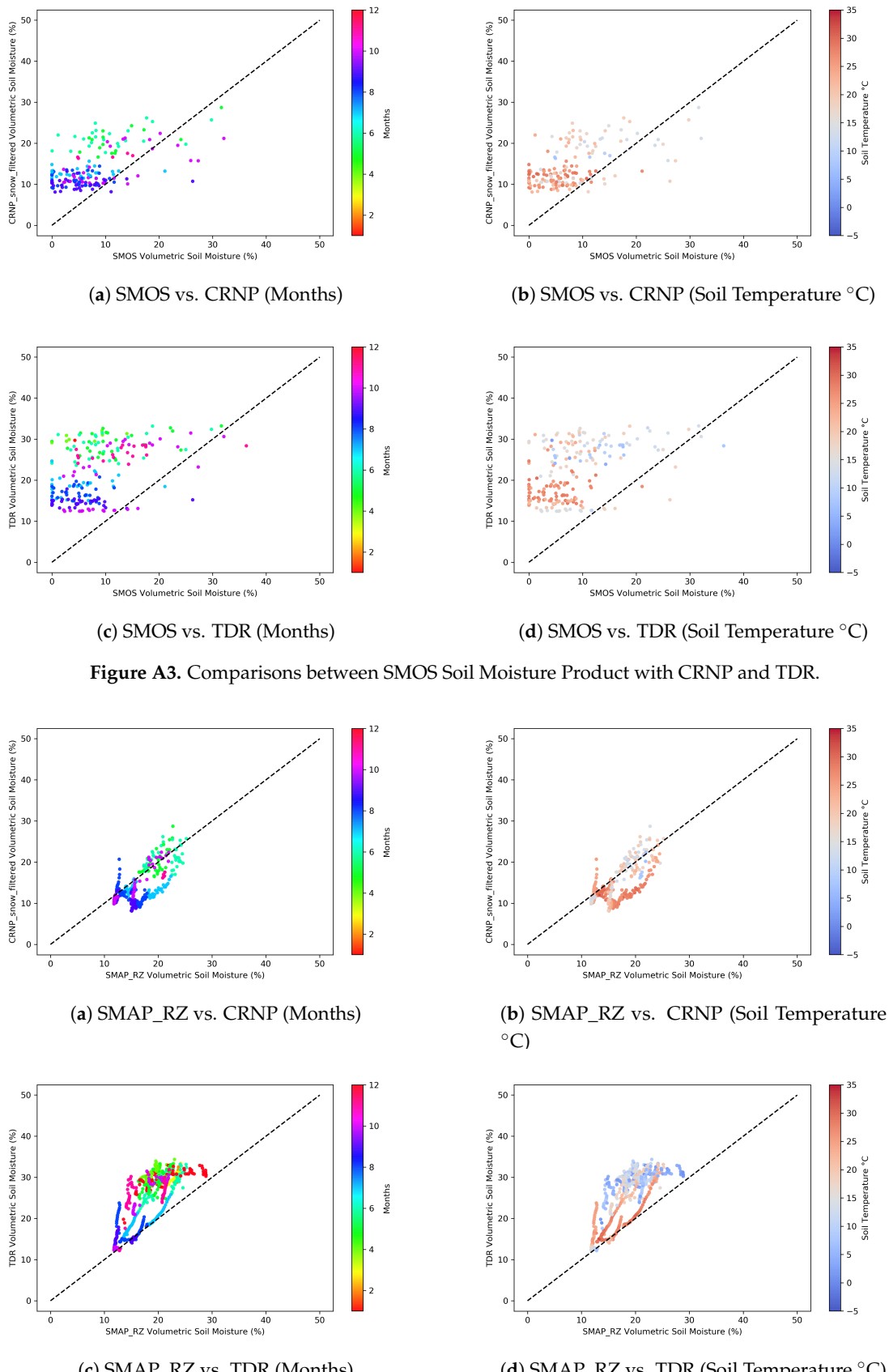

**Figure A3.** Comparisons between SMOS Soil Moisture Product with CRNP and TDR.

**Figure A4.** Comparisons between SMAP_RZ Soil Moisture Product with CRNP and TDR.

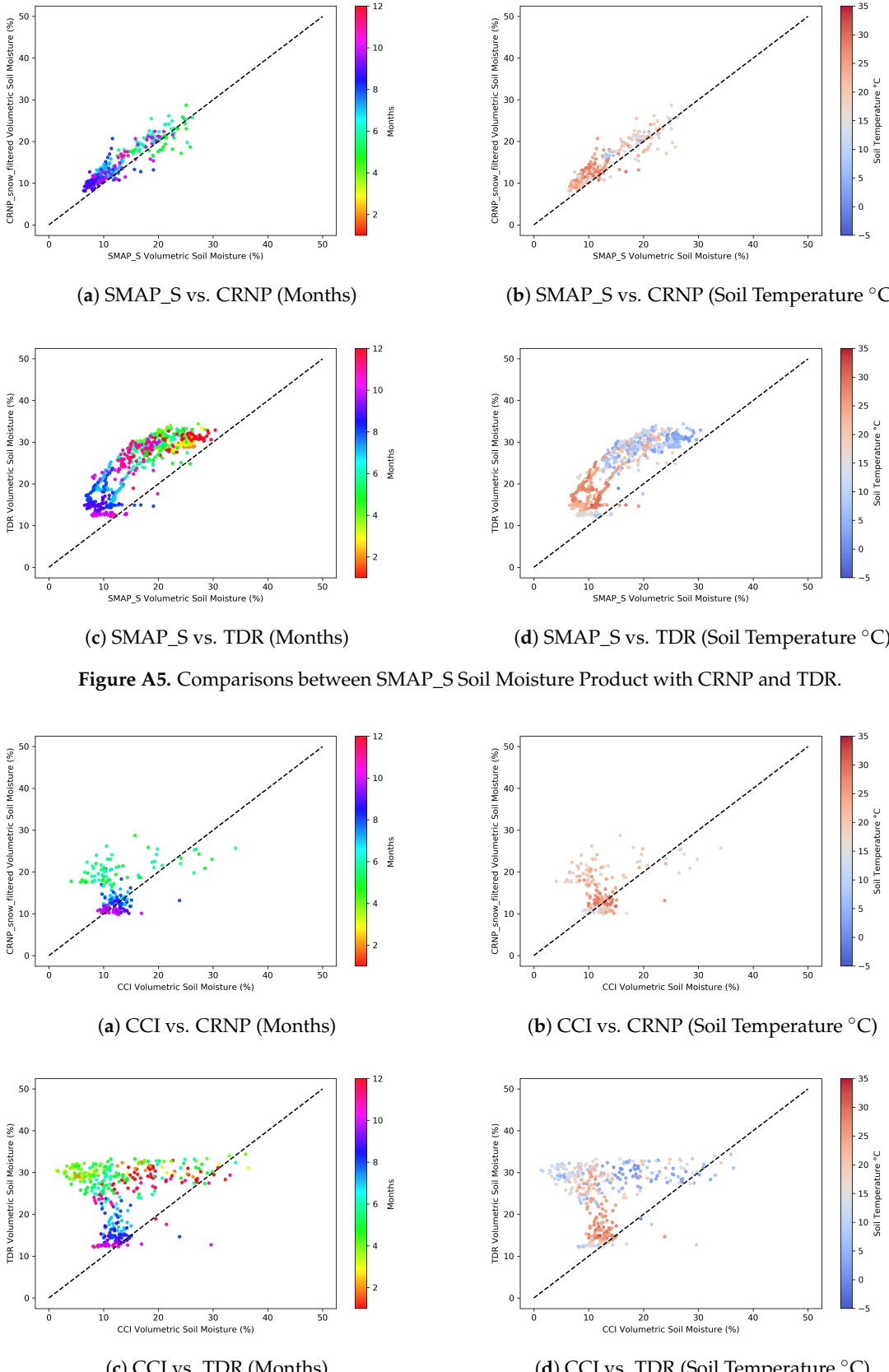

**Figure A5.** Comparisons between SMAP_S Soil Moisture Product with CRNP and TDR.

**Figure A6.** Comparisons between CCI Soil Moisture Product with CRNP and TDR.

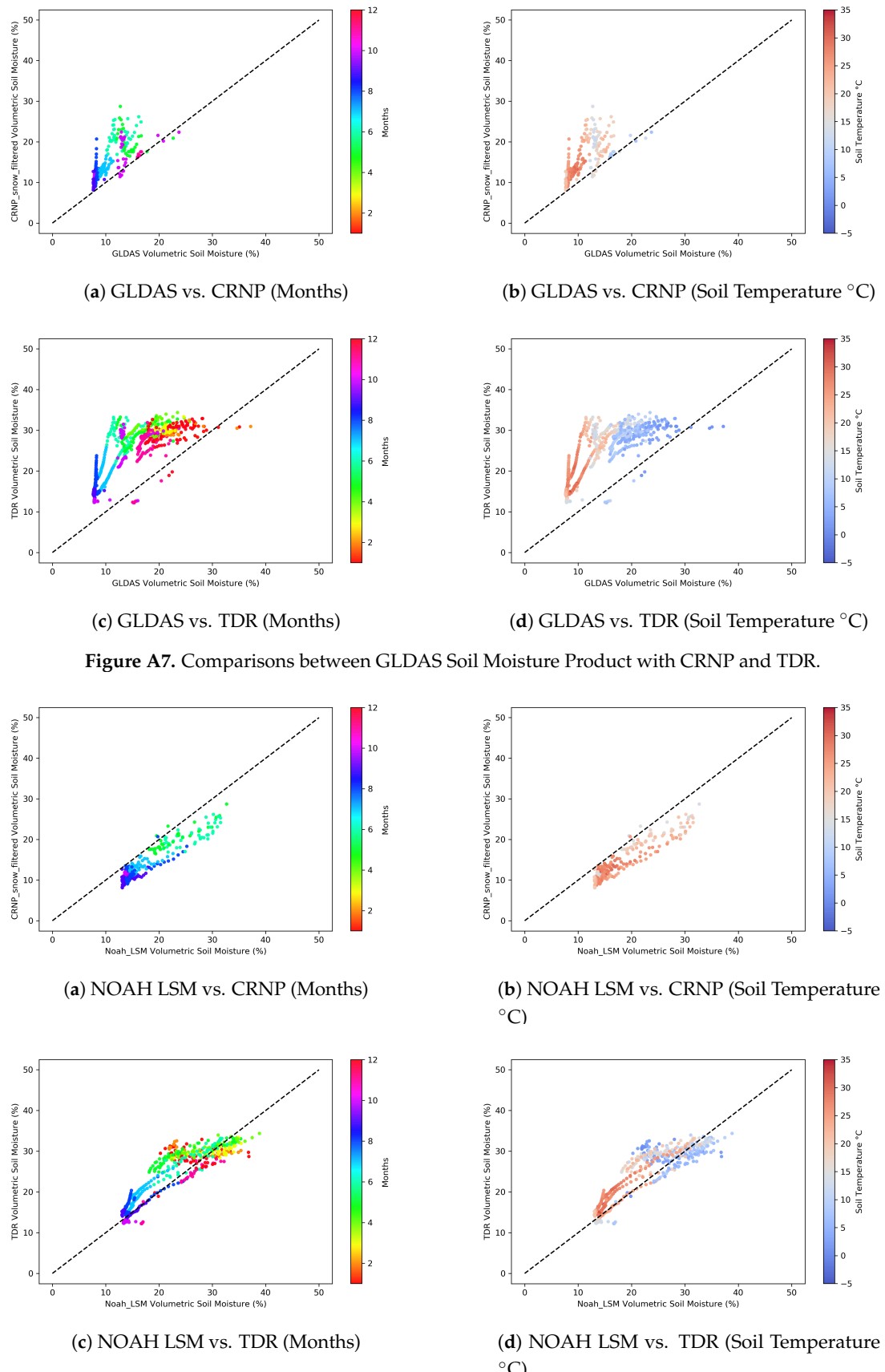

(**a**) GLDAS vs. CRNP (Months)　　　　　　　(**b**) GLDAS vs. CRNP (Soil Temperature °C)

(**c**) GLDAS vs. TDR (Months)　　　　　　　(**d**) GLDAS vs. TDR (Soil Temperature °C)

**Figure A7.** Comparisons between GLDAS Soil Moisture Product with CRNP and TDR.

(**a**) NOAH LSM vs. CRNP (Months)　　　　　(**b**) NOAH LSM vs. CRNP (Soil Temperature °C)

(**c**) NOAH LSM vs. TDR (Months)　　　　　(**d**) NOAH LSM vs. TDR (Soil Temperature °C)

**Figure A8.** Comparisons between NOAH LSM Soil Moisture Product with CRNP and TDR.

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
