# Peer review of "Using Cosmic-Ray Neutron Probes in Validating Satellite Soil Moisture Products and Land Surface Models"

_water, doi:10.3390/w11071362_

Round 1

Reviewer 1 Report

OVERVIEW

The study investigates the use of cosmic-ray neutron probes (CRNP) for the assessment of the quality of satellite and modelled soil moisture products on a global scale and, specifically, at one site in Turkey.

GENERAL COMMENTS

The paper is fairly well written and clear. The topic of the study is relevant and surely of interest for the readership of WATER. The assessment of satellite (and modelled) soil moisture products through CRNP is surely needed as we expect that such technology is more suitable for the validation of coarse resolution products thanks to its larger spatial representativeness. To my knowledge, this is the first study performing a global assessment with the entire COSMOS database, and, hence, it deserves to be published. However, the analysis and the results should be improved before the publication. I listed below the comments to be addressed (in my opinion), with also an indication of their relevance:

1)      MODERATE: The analysis is performed at global scale and at the Turkey site. I understand the reason for such choice, but scientifically there’s no reason for that. The comparison at global scale and at Turkey site should be more similar, see below my suggestions for getting the target.

2)      MAJOR: It is well-known that CRNP accuracy is not as high as TDR or other in situ techniques. Therefore, the assessment through CRNP shouldn’t be considered as a validation. Looking at Figure 7, CRNP has larger errors than SMAP and GLDAS. Therefore, SMAP and GLDAS should be considered as reference! Triple collocation can be effectively used for such purpose, i.e., for the assessment of the quality of three products (assuming that assumptions are met, see my comment below). I strongly suggest to perform the triple collocation analysis also for the 82 sites globally, to have a global assessment of the quality of satellite, modelled, and CRNP data.

3)      MAJOR: Triple collocation analysis performed with different triplets should provide the same results for the same dataset. I mean, in Figure 7 I expect to have the same error for CRNP and NOAH in the different configurations, and it is not the case. Why? Did the authors check the assumptions of triple collocation analysis are met? This step is highly important, particularly if the analysis is carried out on a global scale as suggested above.

4)      MODERATE: The analysis is performed in terms of R and RMSE. However, mainly in soil moisture validation studies, it is much better to use the unbiased RMSE, ubRMSE, to assess the performance as the climatology of soil moisture is not the more important component to be assessed through validation (see e.g., Wagner et al. 2014, doi: 10.1109/TGRS.2013.2282172). Therefore, I suggest using ubRMSE instead of RMSE.

5)      MAJOR: Figures must be improved. In the current version, Figures 6, 8 and 9 are unreadable, and hence not useful for understanding the results.

6)      MAJOR: The description of the results should be improved. Here some suggestions:

a.       Please avoid repetitions. Many times it reads that SMAP is better in semiarid regions and ASCAT in vegetated regions.

b.       In the global assessment, CCI combined shows very good r^2 values (larger than ASCAT) and it is not mentioned at all in the text.

c.       A table showing mean and median values of the performance scores for each product should be added.

d.       In section 3.4.2, too much emphasis to ASCAT soil moisture product is given. It should be balanced with all the products.

e.       In section 4, too much emphasis to SMAP soil moisture product is given. It should be balanced with all the products.

f.        In section 4 the results at Turkey site is described before the global results, whereas in the results the opposite is done, please align them.

In the text I found several errors and typos that should be corrected. I have spotted some of them in the specific comments, a careful rereading is needed.

SPECIFIC COMMENTS (L=line or lines)

L9: Remove “obtained from… (Noah LSM)” is a repetition.

L55: Should be AMSR2. Also correct AMSR with AMSR2 throughout the text.

L80: The morning passes are more successful, not the ascending passes. E.g., SMOS and SMAP have different orbits. It should be corrected throughout the text.

L209: Change “bins” with “triplets”.

L334-335: Note that the areal representativeness of CRNP is still much lower than the footprint of coarse resolution satellite soil moisture products. So the spatial mismatch is still a problem.

Figures 10-17: These figures should be grouped, removed or moved in the supplementary.

L373: Change “great” with “high”.

L381: SMAP (and all satellite soil moisture products) is not able to see soil moisture for snow covered soils. This sentence is wrong.

L402: Check “products” typo

L408: Should be H SAF (without -)

RECOMMENDATION

On this basis, I found the topic of the paper relevant, but I suggest a major revision before the paper can be published on WATER.

Author Response

Response to Reviewer 1 Comments

Point 1: MODERATE: The analysis is performed at global scale and at the Turkey site. I understand the reason for such choice, but scientifically there’s no reason for that. The comparison at global scale and at Turkey site should be more similar, see below my suggestions for getting the target.

Response 1: We have local meteorological and soil moisture data for Çakıt CRNP in Turkey, hence we have selected this site as a case study. The same analyses that have been done for Çakıt CRNP have also been done for all of the stations of the COSMOS database excluding the local Noah LSM model. We try to present the results for one site in order to have the general approach of the analysis more understandable.

Point 2: MAJOR: It is well-known that CRNP accuracy is not as high as TDR or other in situ techniques. Therefore, the assessment through CRNP shouldn’t be considered as a validation. Looking at Figure 7, CRNP has larger errors than SMAP and GLDAS. Therefore, SMAP and GLDAS should be considered as reference! Triple collocation can be effectively used for such purpose, i.e., for the assessment of the quality of three products (assuming that assumptions are met, see my comment below). I strongly suggest to perform the triple collocation analysis also for the 82 sites globally, to have a global assessment of the quality of satellite, modelled, and CRNP data.

Response 2: The triple collocation approach and the related figure (Figure 8) have been changed with respect to the suggestion.

Point 3: MAJOR: Triple collocation analysis performed with different triplets should provide the same results for the same dataset. I mean, in Figure 7 I expect to have the same error for CRNP and NOAH in the different configurations, and it is not the case. Why? Did the authors check the assumptions of triple collocation analysis are met? This step is highly important, particularly if the analysis is carried out on a global scale as suggested above.

Response 3: The triple collocation approach and the related figure (Figure 8) have been changed with respect to the suggestion in point 2. Triple collocation analyses are being conducted by using the methodology provided in (https://pytesmo.readthedocs.io/en/latest/examples.html#triple-collocation-and-triple-collocation-based-scaling) More information about the triple collocation assumptions and the suggested mew approach has been given in the text (Starting from line 327)

Point 4: MODERATE: The analysis is performed in terms of R and RMSE. However, mainly in soil moisture validation studies, it is much better to use the unbiased RMSE, ubRMSE, to assess the performance as the climatology of soil moisture is not the more important component to be assessed through validation (see e.g., Wagner et al. 2014, doi: 10.1109/TGRS.2013.2282172). Therefore, I suggest using ubRMSE instead of RMSE.

Response 4: ubRMSE calculations have been added as suggested (Figure 6.)

Point 5: MAJOR: Figures must be improved. In the current version, Figures 6, 8 and 9 are unreadable, and hence not useful for understanding the results.

Response 5: The figures were made more readable.

Point 6: MAJOR: The description of the results should be improved. Here some suggestions:

a. Please avoid repetitions. Many times it reads that SMAP is better in semiarid regions and ASCAT in vegetated regions.

b. In the global assessment, CCI combined shows very good r^2 values (larger than ASCAT) and it is not mentioned at all in the text.

c. A table showing mean and median values of the performance scores for each product should be added.

d. In section 3.4.2, too much emphasis to ASCAT soil moisture product is given. It should be balanced with all the products.

e. In section 4, too much emphasis to SMAP soil moisture product is given. It should be balanced with all the products.

f.  In section 4 the results at Turkey site is described before the global results, whereas in the results the opposite is done, please align them.

Response 6:

a. Repetition at the introduction part and global assessment results relating SMAP and ASCAT have been removed.

b. CCI has been mentioned in the results.

c. r2, bias, rmse and ubrmse median and mean values have been given in Table 2.

d. In section 3.4.2 the emphasis on the results of ASCAT product has been slightly reduced.

e. In Section 4, authors would like to mention that SMAP is the best product according to validation study and this product can also be used in further practical applications together with CRNP’s. Apart from this information, the same analyses were done for each product and the results are mentioned in the text.

f. Results and conclusions have been aligned as suggested.

Point 7: In the text I found several errors and typos that should be corrected. I have spotted some of them in the specific comments, a careful rereading is needed.

Response 7: Spotted errors and typos have been corrected

Point 8: L9: Remove “obtained from… (Noah LSM)” is a repetition.

Response 8: “obtained from… (Noah LSM)” have been removed

Point 9: L55: Should be AMSR2. Also correct AMSR with AMSR2 throughout the text.

Response 9: AMSR has been corrected with AMSR2 throughout the text

Point 10: L80: The morning passes are more successful, not the ascending passes. E.g., SMOS and SMAP have different orbits. It should be corrected throughout the text.

Response 10: the sentence has been corrected as suggested.

Point 11: L209: Change “bins” with “triplets”.

Response 11: “bins” have been changed with “triplets”.

Point 12: L334-335: Note that the areal representativeness of CRNP is still much lower than the footprint of coarse resolution satellite soil moisture products. So the spatial mismatch is still a problem.

Response 12: The following sentence has been added to Section 3.4.2 (Lines:367-369) “It should be noted that the areal representativeness of CRNP is still much lower than the footprint of coarse resolution satellite soil moisture products, this spatial mismatch effects the correlations between satellite products and CRNP.”

Point 13: Figures 10-17: These figures should be grouped, removed or moved in the supplementary.

Response 13: Figures were moved in the appendix section.

Point 14: L373: Change “great” with “high”

Response 14: “great” has been changed with “high”

Point 15: L381: SMAP (and all satellite soil moisture products) is not able to see soil moisture for snow covered soils. This sentence is wrong.

Response 15: “retreiving soil moisture values even in snow covered durations” has been changed with “providing soil moisture data even in snow covered durations”. SMAP L4 product continuously provides both snow height and soil moisture data which has good correlation with in-situ snow measurements and TDR measurements even at snowy days, however, this issue was not further discussed in the paper since we would like to focus on CRNP and satellite products.

Point 16: L402: Check “products” typo

Response 16: The typo has been corrected.

Point 17: L408: Should be H SAF (without -)

Response 17: H SAF has been corrected.

Author Response

Response to Reviewer 2 Comments

Point 1: In the text there is a repetition in line 9 “obtained from a land surface model (Noah LSM)

Response 1: the repetition “obtained from… (Noah LSM)” have been removed

Point 2: I suggest providing a more complete bibliography including citations about proximal gamma ray spectroscopy sensing, for example:

Strati, V., Albéri, M., Anconelli, S., Baldoncini, M., Bittelli, M., Bottardi, C., Chiarelli, E.,

Fabbri, B., Guidi, V., Raptis, K., Solimando, D., Tomei, F., Villani, G., Mantovani, F., 2018.

Modelling Soil Water Content in a Tomato Field: Proximal Gamma Ray Spectroscopy and

Soil Crop System Models. Agriculture 8, 60. DOI: 10.3390/agriculture8040060

J. Roger Mchenry Angela C. Gill, 1970. Measurement of Soil Moisture with a Portable

Gamma Ray Scintillation Spectrometer. Water Resources Research, Vol. 6, No. 3. DOI:

10.1029/WR006i003p00989

Response 2: proximal gamma ray spectroscopy has been shown as one of the soil moisture detection methods and the bibliography has been improved.

Point 3: The sentence in line 89-90 is not clear, please revise it.

Response 3: The sentence “For this reason, each soil moisture product’s spatial resolution was kept as they are and spatial rescaling was not utilized to them.” Has been changed with “For this reason, satellite products were not spatially or temporally rescaled and used as they are without filling any missing values since filling the missing data by using adjacent grids or time points may affect the analyses adversely.

Point 4: Table 1: please use the degree symbol ° instead of 0 for the coordinate of the pixel center.

Response 4: The symbol has been changed

Point 5: “2.1.5. Climate Change Initiative (CCI)” is not mentioned in the introduction section.

Response 5: CCI has been mentioned in the introduction section in Lines 66-68 as:

“…The aim of this study is to assess the use of CRNP soil moisture data in validation of different types of satellite soil moisture products (active or passive products at ascending and descending nodes), modeled soil moisture as well as a combined soil moisture dataset (CCI)…”

Point 6: Figure 3: I suggest using the same time interval described in line 185 and highlight the data interruption due to a technical problem.

Response 6: Neutron counts data were also added to Figure 3, the data interruption period is so small to be seen in the graph thus it could not be highlighted, it has been given as figure 3b.

Point 7: The definition of RMSE (eq.2) in line 200 is incorrect, please replace it with:                                                

Response 7: The equation has been corrected.

Point 8: Equation 4: seems that a parenthesis is missing.

Response 8: The same equation in bibliography item [34] has been used, there is no more parenthesis.

Point 9: Line 266: why the remaining 22 stations could not be used in the analyses?

Response 9: Timescale of the analyses was taken as March 2015 to December 2018, thus the 22 stations which do not have data in this period could not be used in the analyses. (given in lines 268-272)

Point 10: Figure 4: please align the figures to facilitate the overall understanding.

Response 10: Figures in Figure 4 were aligned as suggested.

Point 11: Figure 5: the quality of the figures is very poor.

Response 11: The quality of Figure 5 is improved, figures were made more readable.

Point 12: Some graphs (e.g. figure 6, 10-17) are too small to properly see the data.

Response 12: Figure 6 is presented to be more readable. Figures were put in the appendices.

Point 13: I encourage the authors to add information about the future perspectives and practical applications.

Response 13: A paragraph has been added to the end of the text (Lines 453-457): “CRNPs show promising potential to be used in conjunction with satellite soil moisture values especially in the field of agriculture. Practical applications of CRNPs and satellite products will help better understanding of the soil moisture deficits in very large fields with lack of a dense soil moisture observation network. In those fields use of CRNPs can lead to better irrigation management and increased yield.”

Round 2

Reviewer 1 Report

The authors have addressed the reviewers' comments and I believe the paper has been greatly improved. However, the structure of the document should be improved (e.g., the figures of the Appendix are in the main text). Moreover, some figures (10-13) are still hard to be read. For these figures, I suggest zooming on the areas in which COSMOS stations are available. 

Author Response

Response : The figures of the Appendix are moved. They are not in the main text any more.

Figures (10-13) are re-prepared and COSMOS stations having large concentration are zoomed in.